# Spinal muscular atrophy-like phenotype in a mouse model of acid ceramidase deficiency

Murtaza S. Nagree [1,2,12], Jitka Rybova[2,12], Annie Kleynerman [2], Carissa J. Ahrenhoerster[2], Jennifer T. Saville[3], TianMeng Xu[4], Maxwell Bachochin[5], William M. McKillop[2], Michael W. Lawlor[6], Alexey V. Pshezhetsky [4], Olena Isaeva [7], Matthew D. Budde[8,9], Maria Fuller [3,10] & Jeffrey A. Medin [1,2,11✉]

Mutations in *ASAH1* have been linked to two allegedly distinct disorders: Farber disease (FD) and spinal muscular atrophy with progressive myoclonic epilepsy (SMA-PME). We have previously reported FD-like phenotypes in mice harboring a single amino acid substitution in acid ceramidase (ACDase), P361R, known to be pathogenic in humans (P361R-Farber). Here we describe a mouse model with an SMA-PME-like phenotype (P361R-SMA). P361R-SMA mice live 2-3-times longer than P361R-Farber mice and have different phenotypes including progressive ataxia and bladder dysfunction, which suggests neurological dysfunction. We found profound demyelination, loss of axons, and altered sphingolipid levels in P361R-SMA spinal cords; severe pathology was restricted to the white matter. Our model can serve as a tool to study the pathological effects of ACDase deficiency on the central nervous system and to evaluate potential therapies for SMA-PME.

[1] Department of Medical Biophysics, University of Toronto, Toronto M5G 1L7 ON, Canada. [2] Department of Pediatrics, Medical College of Wisconsin, Milwaukee, WI 53226, USA. [3] Genetics and Molecular Pathology, SA Pathology at Women's and Children's Hospital, and Adelaide Medical School, University of Adelaide, Adelaide, SA 5006, Australia. [4] CHU Sainte-Justine, Université de Montréal, Montréal, QC H3T 1C5, Canada. [5] University of Wisconsin-Parkside, Kenosha, WI 53144, USA. [6] Department of Pathology and Neuroscience Research Center, Medical College of Wisconsin, Milwaukee, WI 53226, USA. [7] Department of Cell Biology, Neurobiology and Anatomy, Medical College of Wisconsin, Milwaukee, WI 53226, USA. [8] Clement J. Zablocki Veteran's Affairs Medical Center, Milwaukee, WI 53295, USA. [9] Department of Neurosurgery, Medical College of Wisconsin, Milwaukee, WI 53226, USA. [10] Adelaide Medical School, University of Adelaide, Adelaide, SA 5005, Australia. [11] Department of Biochemistry, Medical College of Wisconsin, Milwaukee, WI 53226, USA. [12] These authors contributed equally: Murtaza S. Nagree, Jitka Rybova. ✉email: jmedin@mcw.edu

Acid ceramidase (ACDase) is a lysosomal enzyme encoded by *ASAH1* that catabolizes ceramide into sphingosine and free fatty acid[1,2]. ACDase can also process more complex glycosphingolipids leading to their so-called lyso-, i.e., N-deacylated forms[3]. Recently, it has been shown that ACDase hydrolyzes N-acylethanolamines as well[4]. ACDase deficiency caused by mutations in *ASAH1* leads to Farber disease (FD; OMIM#228000), also known as Farber's lipogranulomatosis, an ultra-rare lysosomal storage disorder (LSD) in which ceramides and related sphingolipids such as sulfatide and GM3 ganglioside accumulate[5,6].

Relatively few FD patients have been described in the literature but a spectrum of disease presentations is evident[7,8]. Patients with the most severe forms of FD die in infancy, while others may survive into adulthood[1,7–9]. Characteristically, patients present with subcutaneous nodules, joint contractures, and a hoarse voice[7,8]. Patients with a more severe form of FD may also suffer from hepatosplenomegaly, along with hematological, respiratory, and neurological pathologies[7,8,10]. Mutations in *ASAH1* have also been associated with spinal muscular atrophy with progressive myoclonic epilepsy (SMA-PME; OMIM#159950)[11]. While caused by mutations in the same gene, SMA-PME has a distinct spectrum of clinical presentations, with an almost exclusive neurological pathology[7,8,11]. SMA-PME patients with a severe disease have progressive motor dysfunction accompanied by worsening chronic myoclonic seizures. Patients succumb to the disorder in late childhood or early adulthood, likely due to respiratory complications[7,8,11–13]. A large spectrum of severity likely exists for this disorder as well, however too few patients have been described to date to accurately determine this[8,11,13–17].

The lack of an association of SMA-PME with ACDase deficiency persisted until 2012, and the rarity of this disorder may have precluded ascertainment of bona fide patients with confirmed pathogenic variants in *ASAH1*[7,8,11]. As such, it is difficult to extrapolate from retrospective studies of historical - and a few more recent - cases describing patients that present with signs strongly suggesting SMA-PME[18–21]. Neurological impairment is evident based on physical examination and electrophysiology of SMA-PME patients[11,15,22,23]. Such clinical signs have, however, often been compared to SMA caused by genetic variants in *SMN1*[11,24,25]. Classical SMA typically involves the peripheral nervous system (PNS) and lower motor neurons of the central nervous system (CNS), with early pathology involving atrophy of anterior horn cells; white matter is typically intact[25–27]. In contrast, a specific area of the CNS or PNS has not been demarcated as commonly and causatively affected in SMA-PME patients. The few studies published for this ultra-rare disorder, to date, are mostly descriptive and primarily pay attention to manifestations, which allude to the heterogeneity of ACDase deficiency in patients.

Guided by SMA-like clinical signs, neuroaxonal degeneration in the spinal cord has been shown in historical cases of SMA coupled with PME[18,19]. Storage pathology in the CNS has also been shown in FD[28,29]. In SMA-PME patients with genetically and biochemically confirmed ACDase deficiency, however, few descriptions of spinal cord pathology are available. One study described a mild reduction in spinal cord volume, as measured by magnetic resonance imaging (MRI)[15]. On the other hand, MRI and other imaging studies have regularly shown negligible and/or mild findings in the brains of SMA-PME patients[11,15,22,23]. White matter pathology, as evident in some other LSDs including Gaucher disease, Niemann-Pick disease type A, GM1 gangliosidosis type 1, and Krabbe disease[30–33], has not been described to date in SMA-PME to our knowledge.

Few mouse models of ACDase deficiency have been generated. Complete elimination of *Asah1* expression resulted in embryonic lethality, with development halting at the 2-cell stage[34,35]. In contrast, we have previously developed and characterized a viable mouse model of FD by introducing a known FD-causing mutation, p.P361R, corresponding to a pathological p.P362R variant in humans, by homologous recombination (P361R-Farber)[36]. These mice are short-lived (8–11 weeks) and present with severe hematopoietic, pulmonary, and hepatic manifestations, as well as clear ophthalmic and CNS phenotypes[36–42]. Phenotypes in P361R-Farber mice, especially storage-laden histiocytic infiltration, seem to closely mimic pathological findings in patients with more severe forms of FD[7,43]. Pathology is accompanied by a massive accumulation of ceramide and some of its derivatives[36–38,40,44]. Another group has developed a viable mouse line in which exon 1 of *Asah1*, which encodes the signal peptide that enables trafficking of the hydrolase to the lysosome, is deleted (Asah1-Δex1)[45]. Despite a seemingly worse mutation, Asah1-Δex1 mice have a phenotype that is largely similar to our P361R-Farber model[36–38,45]. No SMA-PME-like phenotypes in any mouse models of ACDase deficiency have been reported to date.

Our goal in this present study was to create a new mouse model of ACDase deficiency that phenotypically recapitulates signs and known pathology of SMA-PME in patients. Such a mouse model would allow us to shed a spotlight on this understudied disorder to uncover detailed pathological mechanisms associated with ACDase deficiency that exclusively perturb function of the CNS and/or PNS. Uncovering affected tissues/cell types in a unique mouse model of SMA-PME would also facilitate development of targeted diagnostic approaches, as well as treatments, including gene therapies, and provide a valuable resource to test them.

## Results

**Homozygous mice harboring only the T41A mutation in *Asah1* have no detectable phenotype**. A mutation in *ASAH1*, p.T42A, has been reported to cause SMA in patients[14]. We introduced the analogous mutation (T41A) into C57BL/6 mice using Cas9-sgRNA-guided mutagenesis in order to create a unique mouse model of SMA due to ACDase deficiency. We verified carryover of the mutation in *Asah1* transcripts by sequencing *Asah1* cDNA. Homozygous T41A mice occurred at Mendelian ratios ($n = 50$, $\chi 2$-test $p > 0.88$) and were fertile ($n = 3$ each sex). We followed T41A mice for up to 21 months. Subjectively, we did not observe any obvious health issues or changes in activity or behavior. Sphingolipids were largely unchanged in the brain (Supplementary Fig. 1). Only a small decrease in sphingosine was found in the liver and spleen (Supplementary Fig. 1). Importantly, no substantial accumulation of any sphingolipid species was noted (Supplementary Fig. 1), especially relative to P361R-Farber mice[38,40,44].

**Defining the *Asah1* locus in P361R-Farber mice and ACDase activity analyses**. The lack of a phenotypic spectrum amongst Asah1-Δex1, P361R-Farber, and T41A mice suggested that the P361R mutation may not be the sole reason for the severe pathology in our P361R-Farber mice. To confirm the genomic landscape of the *Asah1* locus in P361R-Farber mice, we sequenced a PCR-amplicon derived from exon 12 to 13 and found the NeoR selection cassette still present in intron 12 (Fig. 1a and Supplementary Data 1). We hypothesized that this Pgk-NeoR selection cassette may interfere with *Asah1* gene expression, likely resulting in a hypomorphic allele. This has been observed in other engineered mouse models[46,47]. To test whether intragenic presence of Pgk-NeoR affected the phenotype of P361R-Farber mice we excised it by breeding P361R-Farber mice

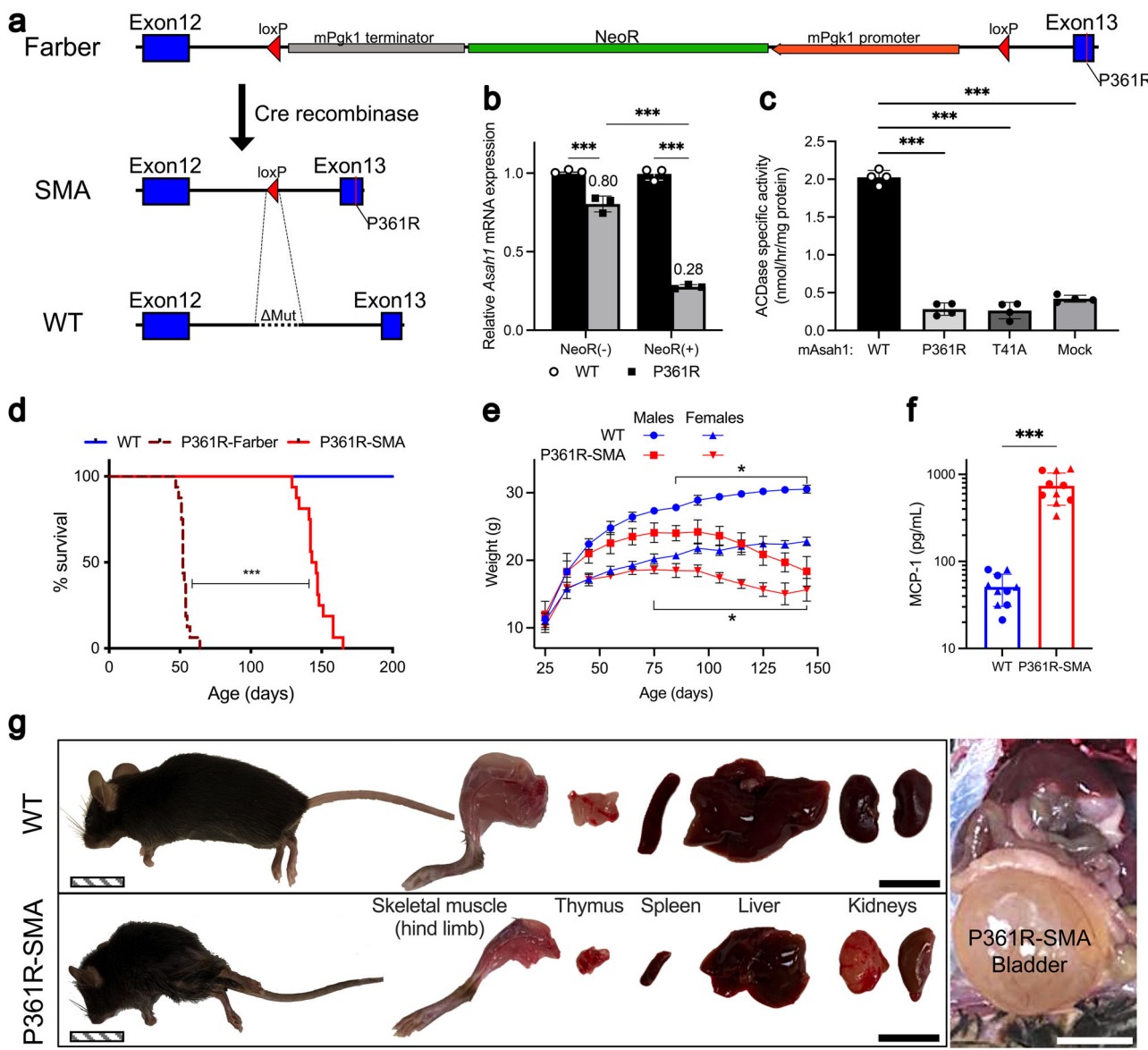

**Fig. 1 The natural history of mutant mice possessing a unique version of the Asah1P361R allele. a** Schematic of the genomic landscape surrounding the SNP leading to P361R before and after excision of the floxed NeoR cassette. The residual loxP site was found to replace 250 bp of intron 12, indicated by ΔMut on the wild-type (WT) schematic. **b** Asah1 mRNA expression in dermal fibroblasts derived from WT and P361R mutant mice with or without the NeoR cassette. Asah1 expression is normalized to matching WT controls and presented as mean relative change ($n = 3$ mice per genotype), with error bars showing standard deviation. Data were compared using a two-way analysis of variance with Tukey's multiple comparisons test, ***$p < 0.001$. **c** Average acid ceramidase activity in lysates of HEK293T cells transfected to overexpress WT, mAsah1-P361R, and mAsah1-T41A ($n = 4$ independent wells); error bars show standard deviation. Activities were compared using a one-way analysis of variance with Tukey's multiple comparisons test, ***$p < 0.001$. **d** Kaplan-Meier survival curve for WT, P361R-Farber, and P361R-SMA mice ($n = 16$ for each genotype). Survival is compared using the Kaplan-Meier estimator (Log-rank test), ***$p < 0.001$. **e** Change in weight over time of male and female WT and P361R-SMA mice ($n = 3$ WT-Male, 5 P361R-SMA-Male, 6 WT-Female and 6 P361R-SMA-Female mice). The same cohort was tracked over time. Mean weights are shown for each time point, error bars show standard deviation. Male and female groups were compared separately using a two-way analysis of variance with Šidák's multiple comparisons tests, *$p < 0.05$. **f** Monocyte chemoattractant protein (MCP)-1 levels in WT and P361R-SMA plasma at 21–23w of age ($n = 10$ biologically independent samples per genotype). Mean levels are shown on a logarithmic Y-axis, error bars show standard deviation, and means compared using a Welch's $t$ test, ***$p < 0.001$. Data points obtained from male and female mice are indicated using circles and triangles, respectively. **g** Organs from WT and P361R-SMA mice, along with a representative image of the distended bladder frequently observed in P361R-SMA mice at ~22w of age. Images were obtained using an Apple iPhone XS Max with default settings. The solid black (hind limb, thymus, spleen, liver and kidney) and white (bladder) scale bars represent 1 cm; the patterned scale bar (mouse) represents 2 cm.

to Sox2-Cre mice. The *Sox2* promoter facilitates germline expression of Cre-recombinase in female heterozygotes irrespective of transgene inheritance in specific oocytes[48]. Offspring from female P361R/+; Sox2-Cre/+ mice were screened for deletion of the NeoR cassette, and the deletion was verified by sequencing a

PCR-amplicon, as above (Fig. 1a). Hereafter these mice are referred to as P361R-SMA mice. We also identified a ~ 250 bp deletion in intron 12 of the P361R allele that was replaced by the residual loxP site when compared to the wild-type (WT) allele (Fig. 1a). As with the T41A mice mentioned above, we confirmed

the presence of the mutation in the *Asah1* transcript by sequencing.

Next, we generated ear skin fibroblasts from P361R-Farber and P361R-SMA mice along with age- and litter-matched WT mice to measure *Asah1* expression. Quantitative PCR showed reduced relative mRNA expression of *Asah1* in fibroblasts from P361R-Farber mice compared to those from P361R-SMA mice (Fig. 1b). Note that we have previously found little or no difference in ACDase activity in tissue extracts from P361R-Farber mice compared to WT or with inhibition of ACDase using carmofur when using a fluorometric assay[49]. To evaluate whether the P361R mutation still retained activity, we transfected HEK293T cells with plasmids that engineered overexpression of mouse WT and mutant ACDase via the CAG promoter. ACDase activity was increased in cell lysates following overexpression of WT ACDase, but not when P361R-ACDase or T41A-ACDase was over-expressed (Fig. 1c). These findings match those previously reported for overexpression of WT and P362R human ACDase[2].

**Partial rescue of a hypomorphic allele in P361R-Farber mice attenuates visceral FD-like phenotypes**. P361R-SMA mice lived longer than the original P361R-Farber mice; median lifespans were 52 days for P361R-Farber mice versus 145 days for P361R-SMA mice ($n = 16$, $p < 0.001$ Log-rank test, Fig. 1d). Younger P361R-SMA mice were fertile and able to produce up to 2 average-sized litters at Mendelian ratios ($n = 4$ each sex, $p > 0.5$ chi-squared test). As with P361R-Farber mice[36], P361R-SMA mice progressively lost weight, though this started much later (Fig. 1e). Similarly, the chemokine MCP-1 was upregulated in plasma of P361R-SMA mice (Fig. 1f), as we previously reported for P361R-Farber mice[36,44,50].

The onset of an observable change in gait in P361R-SMA mice occurred around 15–18 weeks. By 21 weeks mice had kyphosis (Fig. 1g), reduced activity, and tremors. These signs progressed in severity until lower limb paralysis and urinary incontinence occurred (Supplementary Movie 1). Post-mortem examinations revealed that most organs were atrophied in P361R-SMA mice compared to WT mice, including the liver and spleen, a stark contrast to the hepatosplenomegaly observed in P361R-Farber mice[36,44] (Fig. 1g). When incontinence was prolonged and severe, the bladder was severely enlarged and pushed against visceral organs, and asymmetrical kidney damage was observed (Fig. 1g).

To further investigate the pathology of P361R-SMA mice, we examined spleen, liver and bladder by histology (Fig. 2a–c). We found markedly increased Mac-2+ foamy macrophage infiltration into the white pulp (Fig. 2a), but the germinal center structure was relatively conserved compared to WT mice. In the liver we found only perivascular infiltration of Mac-2+ histiocytes (Fig. 2b). In both cases, Lamp1+ staining correlated most with macrophages, though a general increase was observed in P361R-SMA liver parenchyma as well, as would be expected with occurrence of lysosomal storage in those cells (Fig. 2a, b). In the bladders of P361R-SMA mice, we found an increase in cellularity of the urothelium and thickness of the muscle layer compared to WT (Fig. 2c). Bladder wall thickening, lack of evidence for acute inflammation, and sex-independent occurrence of distended bladders strongly suggest neurogenic bladder[51].

Lastly, we compared markers of apoptosis and inflammation in P361R-SMA liver to those from WT mice. As expected, Cathepsin D (CatD), a marker of lysosomal lumen volume, was increased (Fig. 2d, j). Cleaved Caspase 3, a marker of apoptosis, was also slightly increased (Fig. 2e, j). Phosphorylated STAT3 was upregulated but no change in total STAT3 expression was observed (Fig. 2f, g, j). Both total and phosphorylated NF-κB

remained unchanged (Fig. 2h–j). An increase in phosphorylated STAT3 is likely indicative of activated macrophages[52], which is expected to be increased based on results from Mac-2 staining (Fig. 2b).

Together, these data suggest that removal of NeoR generates a new P361R allele by rescuing a potential hypomorphic allele present in P361R-Farber mice. Homozygous mice with this new P361R allele are longer lived and present with predominantly neurogenic phenotypes, with attenuated visceral Farber-like phenotypes.

**Sphingolipid accumulation in the spleen and liver of P361R-SMA mice**. We and others have previously demonstrated massive accumulation of ceramide, and substantial secondary elevation of some other sphingolipids, in various organs from ACDase-deficient mice[36,38,40,44,45]. We first quantified sphingolipids (Fig. 3a) in tissue extracts from the spleen and liver of P361R-SMA mice. Sphinganine (Spn) was increased in both tissues (Fig. 3b, m). Dihydroceramide (dhCer) (Fig. 3c, n) was also increased in addition to ceramide (Cer; Fig. 3d, o). Elevations were more dramatic in the spleen (~30-fold) than in the liver (~4-fold), but overall were less increased compared to previously reported changes in P361R-Farber mice (>75-fold in spleen and ~20-fold in liver)[36,38,44]. This corroborates our observation of an attenuated FD-like phenotype. Sphingosine (Sph), on the other hand, was significantly increased in both tissues (Fig. 3e, p). We also confirmed a trend towards an increase in Spn (Supplementary Fig. 2a, b), dhCer (Supplementary Fig. 2d, e), and Sph (Supplementary Fig. 2g, h) in extracts from the spleen and liver of P361R-Farber mice used as controls for this experiment, which suggests such elevations are not specific to P361R-SMA mice.

We also found significant but variable elevation of ceramide-1-phosphate (C1P) in spleen and liver (Fig. 3f, q), and secondary elevation of more complex sphingolipids (Fig. 3g–k, r–v). Relative abundance of acyl-chain variants (ACVs) of the various sphingolipid species measured here were calculated when possible (Fig. 3l, w); we were unable to calculate these for trihexosylcer-amides (THC) as many ACVs were not detected in WT tissue. The most consistent shift in abundance was an increase in C16:0 (Fig. 3l, w). Increases in ACVs, especially of dhCer and Cer, were more biased in liver extracts—towards C16:0 especially—than in spleen (Supplementary Data 2). Note that we have previously reported similar trends for liver in P361R-Farber mice[38].

**P361R-SMA mice have neurological indications that are not caused by brain pathology**. P361R-SMA mice displayed a pronounced neurological phenotype, with progressive deficits in motor function (Supplementary Movie 1). We first evaluated skeletal muscle histopathology in several mice. Comparison of similarly oriented areas of muscle identified a subpopulation of myofibers in P361R-SMA that were smaller than the corresponding population in WT mice, suggesting mild myofiber atrophy (Supplementary Fig. 3), although it should be noted that sampling error may be responsible for this apparent difference.

To quantify changes in sensory-motor function over time, we conducted behavior tests on P361R-SMA mice. The wire-hang test was used to assess neuromuscular strength at three time points. Motor deficits were measurable as early as 9-10 weeks of age in a subset of mice ($p = 0.05$, WT vs. P361R-SMA) and worsened with age (Supplementary Fig. 4a). P361R-SMA mice were unable to perform this test by 19-20 weeks. We used the rotarod test to evaluate motor coordination at similar time points. In contrast to the dramatic reduction in hang time with the wire-hang test, P361R-SMA mice performed well at younger ages on the rotarod test (Supplementary Fig. 4b). Performance was only significantly reduced at 19-20 weeks, though mice were still able

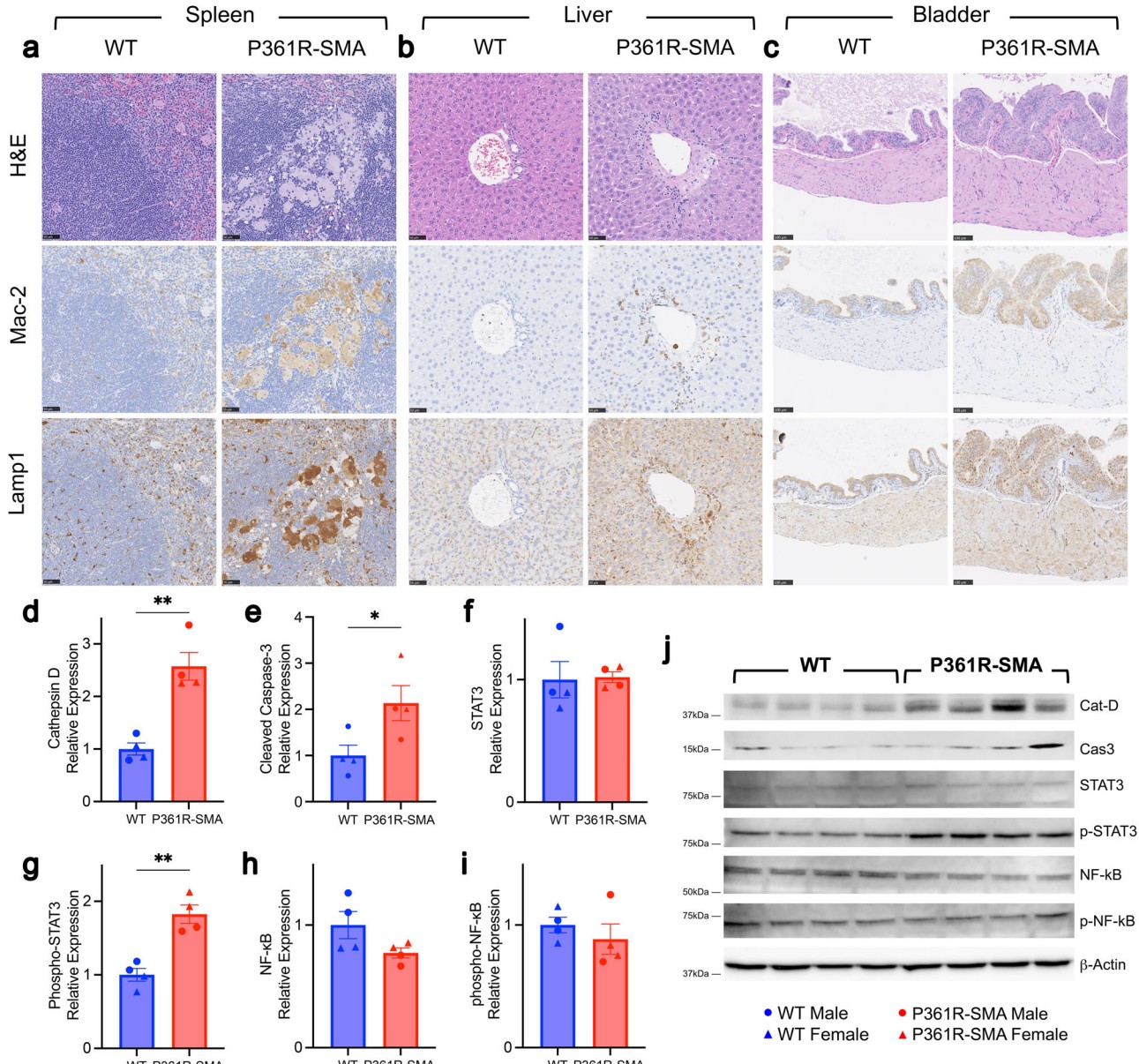

**Fig. 2 Histological examination of peripheral tissues from P361R-SMA mice.** Sections from spleens (**a**), livers (**b**) and bladders (**c**) of wild-type (WT) and P361R-SMA mice were stained using hematoxylin and eosin (H&E), or immunostained for macrophages (Mac-2) or lysosomes (Lamp1). Scale bars represent 50 μm in (**a**) and (**b**), and 100 μm in (**c**). WT and P361R-SMA liver homogenates were Western blotted, and densitometry was used to quantify Cathepsin D (**d**), cleaved Caspase 3 (**e**), total STAT3 (**f**), phosphorylated STAT3 (**g**), total NF-κB (**h**), and phosphorylated NF-κB (**i**) from the representative immunoblots. **j** Relative ratios are plotted, obtained by deriving band densities for the major expected band and normalized first to β-Actin, then to average density in WT. Bars represent mean values ($n = 4$ mice per genotype), error bars depict standard error of the mean, and points show measurements for individual mice where triangles are females and circles are males. Data were compared using Welch's $t$ test, $^{**}p < 0.005$, $^*p < 0.05$.

to maintain balance for some time (Supplementary Fig. 4b). These data suggest motor function in P361R-SMA mice is affected to a greater degree than coordination. We then used sensitivity to von Frey filaments to examine P361R-SMA mice for mechanical sensory defects over time. Paw-withdrawal responses to a 1.0 g filament were significantly reduced in 18- and 21-week-old P361R-SMA mice compared to age-matched WT controls (Supplementary Fig. 4c). Similarly, the amount of force needed to elicit a paw-withdrawal response was increased in 18- and 21-week-old P361R-SMA mice (Supplementary Fig. 4d).

We used the flurothyl model of repeated generalized seizure to determine whether P361R-SMA mice exhibit increased susceptibility to seizure development. Fifteen-week-old P361R-SMA mice and age-

matched WT animals were subjected to eight flurothyl-induced seizures (once a day). Susceptibility to seizure generation was evaluated during the first and eighth days of exposure to flurothyl (induction phase; Supplementary Fig. 4e, f) and after a 28-day rest period (rechallenge phase; Supplementary Fig. 4g–j). All mice developed generalized seizures in response to flurothyl exposure. We did not find a significant difference in the latency and duration of generalized seizures during the training (Supplementary Fig. 4e, f) or rechallenge phases of the flurothyl model (Supplementary Fig. 4i, j). Myoclonic jerks of the head and neck musculature, expressed before the onset of generalized seizures, are often observed during flurothyl exposure[53,54]. Only a few mice displayed myoclonic jerks during the training phase. The lack of myoclonic jerks upon acute seizure

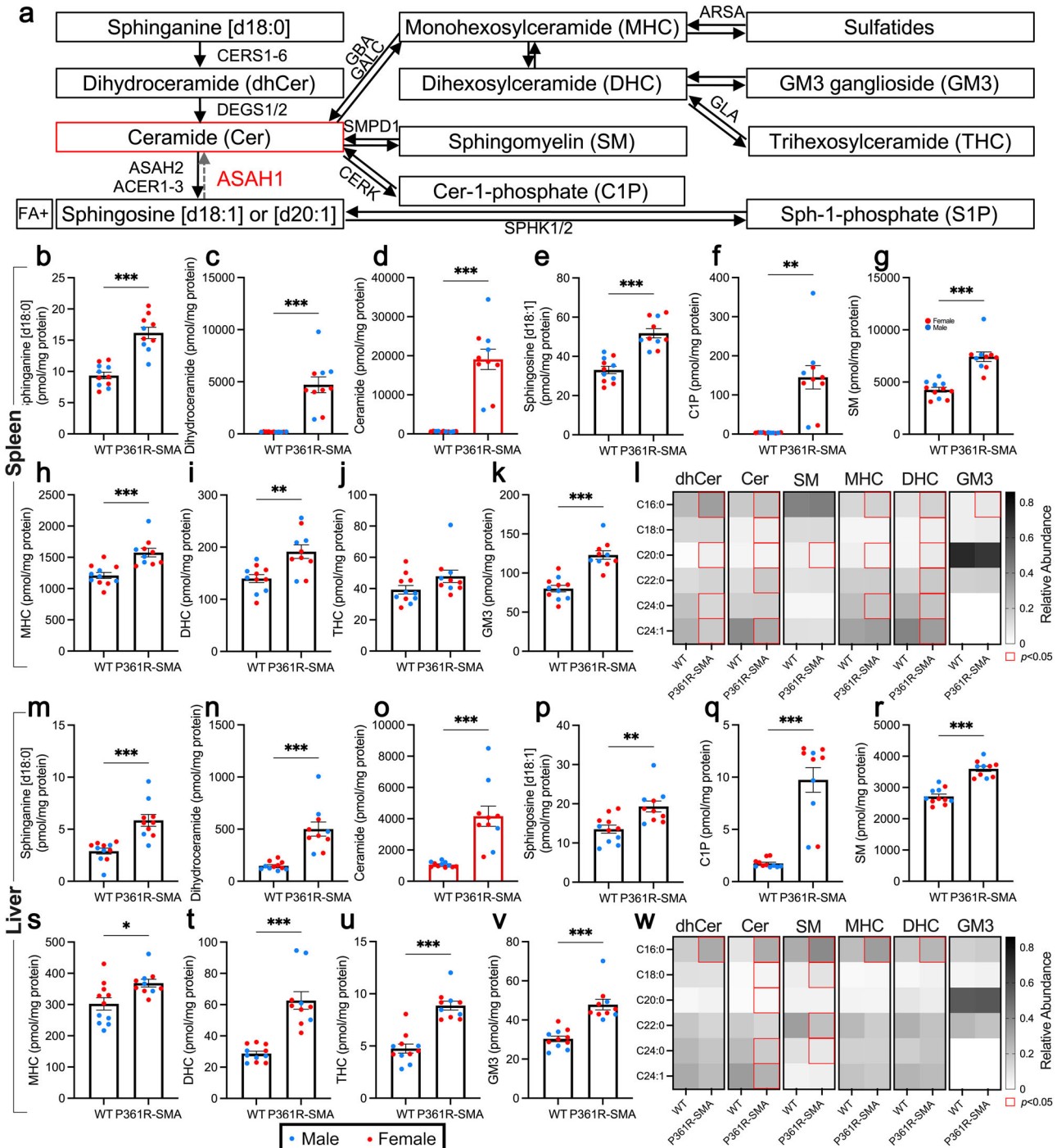

induction during the first challenge prompted us to adopt the induction-rechallenge model. Accordingly, all mice expressed myoclonic jerks in response to flurothyl exposure during the rechallenge phase. We found a significant decrease in the flurothyl-induced myoclonic jerk threshold (Supplementary Fig. 4g) and a trend to an increased frequency of myoclonic jerks in P361R-SMA mice (Supplementary Fig. 4h). Together, these data indicate the presence of a complex sensory-motor deficit in P361R-SMA mice, with skeletal muscle weakness having earlier onset.

Other mouse models of LSDs show sensory-motor dysfunction due to cerebral and/or cerebellar pathology, including gross destruction of cerebellar structure, demyelination, and patterned loss of Purkinje cells[49,55–57]. To examine this in P361R-SMA mice, we stained cerebella with Luxol Fast Blue (LFB) to look for

these facets of neuropathology (Supplementary Fig. 4k). We did not find any gross changes in structure, and only observed mild and inconsistent demyelinated areas (Supplementary Fig. 4k). We also did not find any loss of Purkinje cells, nor did we find distention of Purkinje cells (Supplementary Fig. 4k) previously observed at a mild level in P361R-Farber mice[40]. We further examined cerebella and motor cortices for GFAP+ activated astrocytes and for the presence of CD68+ microglia/macrophages by immunofluorescence (Supplementary Fig. 4l–n). GFAP+ staining was found in abundance throughout the cerebella of P361R-SMA mice, and although mild, was still significantly elevated in their motor cortices (Supplementary Fig. 4l, n). CD68+ staining was limited to the *arbor vitae* in P361R-SMA cerebella, and was less pronounced than GFAP+ staining

**Fig. 3 Ceramides and related sphingolipids in extracts from the spleens and livers of 21-week-old P361R-SMA mice. a** Summary of a portion of sphingolipid metabolic pathways with gene names for select enzymes. Corresponding enzyme names are listed below. Shorthand notation for some sphingolipid families are also shown. Arrows show potential for interconversion. Note that reverse activity of ceramidases utilizing sphingosine has only been shown in vitro. **b–k** Levels of the indicated sphingolipids were measured in spleen homogenates by mass spectrometry. Ceramide data is highlighted with red bars. Total levels were calculated by adding up molar amounts of individually measured acyl-chain variants. Values for individual mice are shown as dots (wild-type (WT), $n = 11$ mice; P361R-SMA, $n = 10$ mice), sex is differentiated using different colors (blue for male and red for female), bars throughout represent the mean and error bars show standard error of the mean. Total levels were compared using a Welch's $t$ test, ***$p < 0.001$, **$p < 0.01$ and *$p < 0.05$. Sphingosine [d20:1], sphingosine-1-phosphate, and sulfatides were not detected. **l** Heatmap of the relative abundance in spleen extracts of selected acyl-chain variants of the indicated sphingolipid families. Data were compared using a two-way analysis of variance with Šidák's multiple comparison test; statistically significant ($p < 0.05$) differences are indicated with a red outline. **m–v** Sphingolipid levels, as specified, were assessed in liver homogenates. Data is presented and compared as described for (**b–k**). **w** Relative abundance of select acyl-chain variants of the specified sphingolipid families in liver extracts were calculated and are displayed as a heatmap. Data was compared, and statistical significance displayed, as described for (**l**). The gene names in (**a**) correspond to the following enzymes: CERS1-6 ceramide synthases 1 through 6, DEGS1/2 dihydroceramide desaturase 1 and 2, ASAH1 acid ceramidase, ASAH2 neutral ceramidase, ACER1-3 alkaline ceramidases 1 through 3, GBA β-glucocerebrosidase, GALC galactosylceramidase, SPMD1 acid sphingomyelinase, CERK ceramide kinase, SPHK1/2 sphingosine kinase 1 and 2, ARSA arylsulfatase A, GLA α-galactosidase A.

(Supplementary Fig. 4m, n). The motor cortex did not have appreciable CD68+ staining (Supplementary Fig. 4n). These data suggest that, while some level of demyelination and injury is present in the brain, the magnitude of change is unlikely to result in the severe neuronopathic phenotype observed in P361R-SMA mice (Supplementary Movie 1 and Supplementary Fig. 4a–d).

**Sphingolipid accumulation in P361R-SMA mouse brains.** We next measured sphingolipid levels in brain lysates from P361R-SMA mice to determine any biochemical impact of ACDase deficiency, and to compare with previously reported findings in P361R-Farber mice[40]. Spn levels were decreased in P361R-SMA mice compared to WT (Supplementary Fig. 5a). Total dhCer and Cer levels were increased in tissue lysates from P361R-SMA mice (Supplementary Fig. 5b, c), but changes were smaller than those previously reported in P361R-Farber mice[40]. Two Sph species, [d18:1] and [d20:1], were detected; [d18:1] Sph was found to be somewhat reduced in P361R-SMA mice (Supplementary Fig. 5d, e). Sph-1-phosphate (S1P) was also detected in brain lysates and found to be significantly reduced in extracts from P361R-SMA mice (Supplementary Fig. 5f). Changes in Spn and Sph (Supplementary Fig. 5a, d) levels contrasted with our previously published findings where Spn levels were unchanged and Sph elevated[40]. Spn and Sph, as well as dhCer, in lysates from control P361R-Farber brains trended in the same direction as the P361R-SMA brains (Supplementary Fig. 2c, f, i). Again, this suggests these changes are not unique to P361R-SMA mice. As postulated for liver and spleen, the differences between measurements in this study versus those previously published may arise due to differences in genetic background.

Total levels of C1P, sphingomyelin (SM), monohexosylceramide (MHC), and dihexosylceramide (DHC) were not significantly different between lysates of brain tissues derived from P361R-SMA and those from WT mice (Supplementary Fig. 5g–j). Total trihexosylceramide (THC) was significantly elevated in P361R-SMA mice, but concentrations remained relatively low (Supplementary Fig. 5l). Monosialodihexosylganglioside (GM3) was slightly but significantly elevated (Supplementary Fig. 5l). There was no change in sulfatide (Supplementary Fig. 5m). Despite minimal changes in the total concentrations, the proportion of ACVs were significantly different. A general increase in C16:0 ACVs was seen, as well as longer ACVs, at the expense of C18:0 ACVs (Supplementary Fig. 5n). We have previously reported similar changes in relative ACV abundance in brain extracts from P361R-Farber mice[40]. Surprisingly, we found no differences in C18:0 Cer in brain extracts from P361R-SMA mice compared to WT, and a significant but relatively small change in C18:0 dhCer (Supplementary Data 2). Reduced relative abundance of the C18:0 Cer and dhCer ACVs (Supplementary

Fig. 5n) is therefore a result of accumulation of other ACVs and not due to reduced C18:0 Cer and dhCer concentrations. Additionally, despite little to no change in total levels of the more complex sphingolipids (Supplementary Fig. 5h–m), significant changes in individual ACVs were found (Supplementary Data 2). Shorter and longer SM ACVs were increased, while mid-length ACVs were decreased (Supplementary Data 2). MHC were detected at high concentrations in extracts from brains of P361R-SMA mice. MHC are comprised of both glucosyl-(GlcCer) and galactosylceramide (GalCer), which have distinct anabolic pathways. GlcCer is the precursor for DHC, whereas GalCer is sulfated, producing sulfatides (Fig. 3a)[58]. Comparisons of molar amounts of MHC and sulfatides (Supplementary Fig. 5i, m, Supplementary Data 2), and the relative abundance of their ACVs (Supplementary Fig. 5n, Supplementary Data 2) suggests that the majority of MHC is indeed GalCer, as expected[59,60]. While total MHC remained unchanged (Supplementary Fig. 5i), the C16:0 isoform was slightly but significantly elevated (Supplementary Data 2), as reported previously for P361R-Farber mice[40]. Surprisingly, significant reductions in longer MHC ACVs were found, particularly in C23:0 (Supplementary Data 2).

ACV-specific changes of sulfatide in tissue extracts from P361R-SMA mice mirrored those of MHC, where no significant elevations in total concentration was observed, but reduced longer ACVs, especially C23:0 and C23:0(OH) sulfatides, were noted (Supplementary Data 2). A reduction in this sulfatide species in cerebrospinal fluid has been previously associated with progressive multiple sclerosis, although this was accompanied by significant increases in other ACVs[61]. We have also previously reported elevations in all measured DHC ACVs in P361R-Farber mice, as well as elevation in C18:0 GM3[40]. Here we find significant elevations in all DHC ACVs measured in brain extracts from P361R-SMA mice except for C20:0, which was not significantly changed, and C18:0, which was significantly reduced (Supplementary Data 2). Elevations in DHC ACVs in P361R-SMA mice compared to WT were also smaller than those previously reported for P361R-Farber mice[40]. Similarly, while we found a slight but significant increase in C16:0 GM3, the C18:0 ACV was unchanged (Supplementary Data 2), further highlighting the distinct disease states found in P361R-Farber versus P361R-SMA mice.

**P361R-SMA mice have a severe and progressive spinal cord pathology.** Given a relatively mild brain pathology, we hypothesized involvement of the spinal cord. Ex vivo T2-weighted MRI of fixed vertebral columns from our P361R-SMA mice showed widespread white matter hyperintensities (Fig. 4a), which may be indicative of inflammatory, demyelinated lesions[40,62,63].

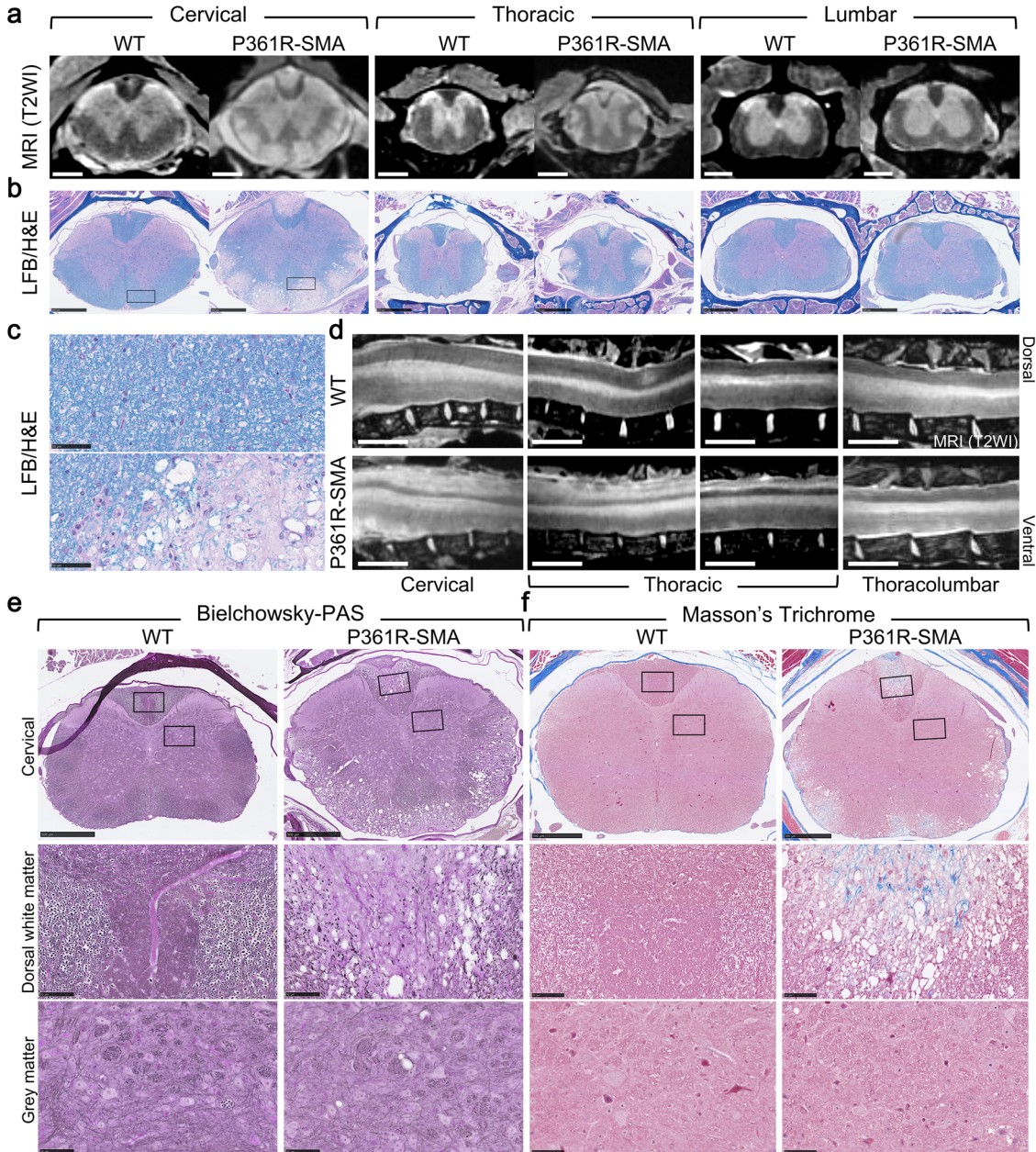

**Fig. 4 Ex vivo MRI and histological staining of spinal cords of 21-23-week-old P361R-SMA mice. a** Formalin-fixed spinal columns from wild-type (WT) and P361R-SMA mice were imaged using T2-weighted MRI (T2WI). Selected axial slices are shown from cervical, thoracic, and lumbar regions. The scale bar represents 500 μm. **b–c** Histological follow-up to evaluate myelination in spinal cord sections from (**a**) was carried out using luxol-fast blue and hematoxylin & eosin staining (LFB/H&E) (**b**). The scale bar represents 500 μm. A higher magnification of the regions indicated by a box are shown in (**c**); the scale bar represents 50 μm. **d** Sagittal views of spinal column MRIs focused on the dorsal white matter tract. The scale bar represents 1000 μm. **e** Bielchowsky staining of spinal cord sections with Periodic acid-Schiff (PAS) counterstain. A higher magnification of the boxed regions, representative of white (middle) and gray (bottom) matter are shown. Axonal projections are stained black/dark gray. The scale bars represent 500 μm for the cervical panel, and 50 μm for the dorsal white matter and the gray matter-focused panels. **f** Masson's trichrome staining of spinal cord sections. A higher magnification of the boxed regions, representative of white (middle) and gray (bottom) matter are shown. Blue staining indicates fibrotic deposits (collagen). The scale bars represent 500 μm for the cervical panel, and 50 μm for the dorsal white matter and the gray matter-focused panels.

Indeed, follow-up examination of spinal cords using LFB confirmed demyelination and showed vacuolization (Fig. 4b, c). Pronounced white matter lesions were present throughout the ventral and lateral columns, continuously in the dorsal column, and reduced in severity in the thoracolumbar region and below (Fig. 4a–d).

Given the progressive nature of the sensory-motor dysfunction observed (Supplementary Fig. 4a–d), we hypothesized the severity of spinal cord pathology would follow a similar trajectory. To investigate this, we stained spinal cord sections from 8–9 week-old (Supplementary Fig. 6a, b), and 15 week-old (Supplementary Fig. 6c–e) P361R-SMA mice with LFB. Lesions progressively worsened, and demyelination started as early as 8 weeks, despite a lack of an observable motor deficit or change in weight at that age. Further, closer examination of the lesion in the dorsal funiculus at 15 weeks (Supplementary Fig. 6e) showed intact

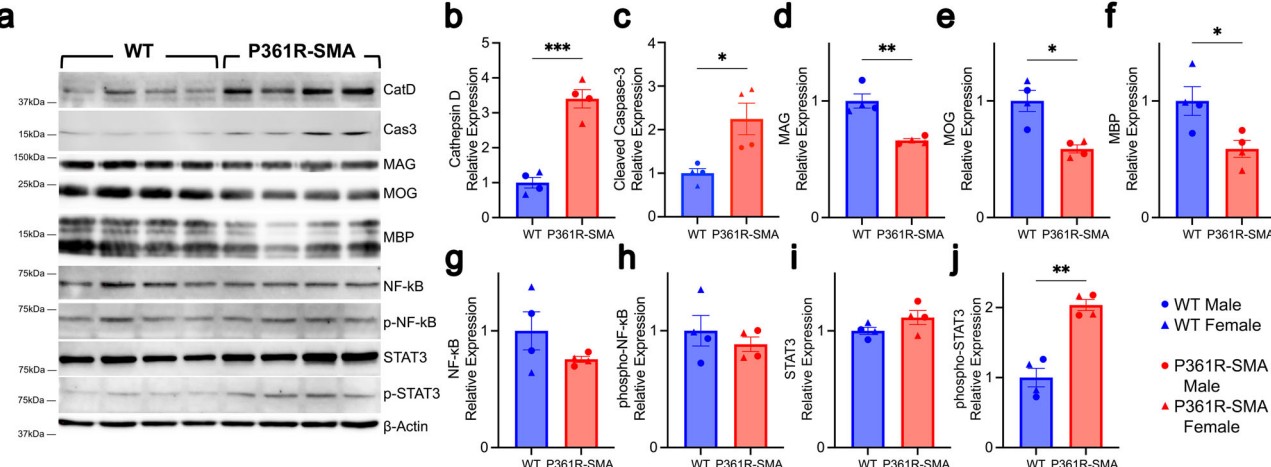

**Fig. 5 Western blotting for markers of cell death, inflammation and myelination in spinal cord lysates from 21-week-old P361R-SMA mice.**
**a** Representative Western blots of wild-type (WT) and P361R-SMA spinal cord lysates for the indicated proteins. **b–j** Densitometry of the Western blots in (**a**) for Cathepsin D (CatD) (**b**), cleaved Caspase 3 (Cas3) (**c**), myelin binding protein (MBP); (**d**), myelin-associated glycoprotein (MAG); (**e**), myelin oligodendrocyte glycoprotein (MOG); (**f**), total STAT3 (**g**) and phosphorylated STAT3 (p-STAT3), (**h**), and total NF-κB (**i**) and phosphorylated NF-κB (p-NF-κB), (**j**). Band densities were calculated for the major expected band and normalized first to β-Actin, then to average density in WT. Bars represent mean values ($n = 4$ mice per genotype), error bars depict standard error of the mean, and points show measurements for individual mice where triangles are females and circles are males. Data were compared using Welch's *t* test, \*\*\*$p < 0.001$, \*\*$p < 0.01$ and \*$p < 0.05$.

structure of some regions, whereas most structure appeared to be destroyed when mice were older than 21 weeks (Fig. 4b). In contrast to the remarkable white matter pathology in P361R-SMA mice, none of the cases we examined had any appreciable disruption of gray matter structure (Fig. 4b). LFB staining of spinal cords from moribund P361R-Farber mice revealed only mild demyelination (Supplementary Fig. 6f).

To investigate the integrity of axons, we silver-stained spinal cord sections from our P361R-SMA mice. A prominent loss of silver staining was seen throughout white matter lesions, especially in the dorsal funiculus (Fig. 4e), strongly suggesting loss of axonal processes. Loss of silver staining appeared to be less severe compared to the extent of demyelination (Fig. 4b, e); which was especially noticeable in the lateral and ventral funiculi. Loss of myelin has been suggested to precede loss of axons in other disorders[64–69]. Periodic acid-Schiff (PAS) counter-staining revealed abnormal scar-like structures in lesions (Fig. 4e). To follow up, we stained sections with Masson's trichrome. Fibrotic scarring, usually indicated by abnormal presence of blue-colored connective tissue, was present within lesions (Fig. 4f). Silver staining also confirmed the integrity of axons in gray matter (Fig. 4e) and this, along with the lack of any connective material (Fig. 4f), suggests that the gray matter was normal in the regions examined.

**Cell death and inflammatory cell infiltration in P361R-SMA spinal cords.** Fibrotic scars observed in lesions (Fig. 4f) can be an indicator of glial cell activation and/or infiltration of immune cells, as described in models of other demyelinating disorders and in spinal cord injury[70–72]. We observed a general increase in CatD content in P361R-SMA mice by Western blotting of whole spinal cord lysates (Fig. 5a, b), confirming an expected increase in lysosomal lumen. A concomitant increase in cleaved-Caspase 3 was also found (Fig. 5a, c), suggesting apoptotic cells. To confirm a broad loss of myelin, we also probed for myelin- and oligodendrocyte-associated proteins, which we found to be reduced (Fig. 5a, d–f), suggesting a good fraction of the apoptosis we observed is that of oligodendrocytes[73]. To provide insight into inflammatory pathway activation, we probed for STAT3 and NF-κB as with P361R-Farber[38] and P361R-SMA livers (Fig. 2f–j).

p-STAT3 was upregulated in spinal cords but p-NF-κB remained unchanged (Fig. 5g–j).

Astrocyte activation and/or macrophage recruitment are both characterized by upregulated p-STAT3 in the CNS[74,75]. We first verified loss of oligodendrocytes/myelin and axons in lesions in white matter by staining for CNPase and βIII-Tubulin, respectively (Fig. 6a, b). Myelinated axons were replaced by cells with high CatD staining, implying lysosomal storage. This was especially obvious in the dorsal funiculus (Fig. 6a, b). We then stained for GFAP to label astrocytes. Here we saw a large increase in GFAP+ staining in the white matter of P361R-SMA spinal cords (Fig. 6c), as also observed in P361R-SMA cerebella sections (Supplementary Fig. 4l). GFAP staining overlapped partially with CatD staining but did not appear to be associated with larger clusters (Fig. 6c).

We and others have previously shown association of dense CatD/Lamp1+ lysosomal lumen with Mac-2+ staining in various tissues in P361R-Farber mice[38,41,45] including optic nerves[41]. We observed the same in liver and spleen of P361R-SMA mice (Fig. 2a, b). We thus stained serial sections of spinal cord with LFB, and for Mac-2 and CatD. We found prominent Mac-2 staining corresponding to the location of lesions, especially in the dorsal funiculus (Fig. 6d), suggesting infiltration of macrophages. Sparse macrophages were found in which LFB staining was present, indicating active clearance of myelin debris (Fig. 6d, inset). Some densely stained Mac-2+ cells that were morphologically similar to microglia were also seen in the gray matter (Fig. 6d). Large vacuolar structures, however, remained unstained with either Mac-2 or CatD (Fig. 6d). Vacuolar structures may be an artifact of tissue processing but are more likely to reflect late-stage spongy degeneration with edema, as occurs in Canavan disease, for example[76]. Hyperintensities observed in white matter by MRI (Fig. 4a) support the latter conclusion.

In 8-week-old mutant mice, we confirmed the minor loss of myelin seen with LFB staining (Supplementary Fig. 6a) by immunostaining (Supplementary Fig. 7a). We found slightly perturbed axonal staining (Supplementary Fig. 7b) in P361R-SMA mutant mice where gaps in staining were occupied by CatD-positivity. GFAP-stained cells were present in lesions, some of which clearly overlapped with CatD (Supplementary Fig. 7c,

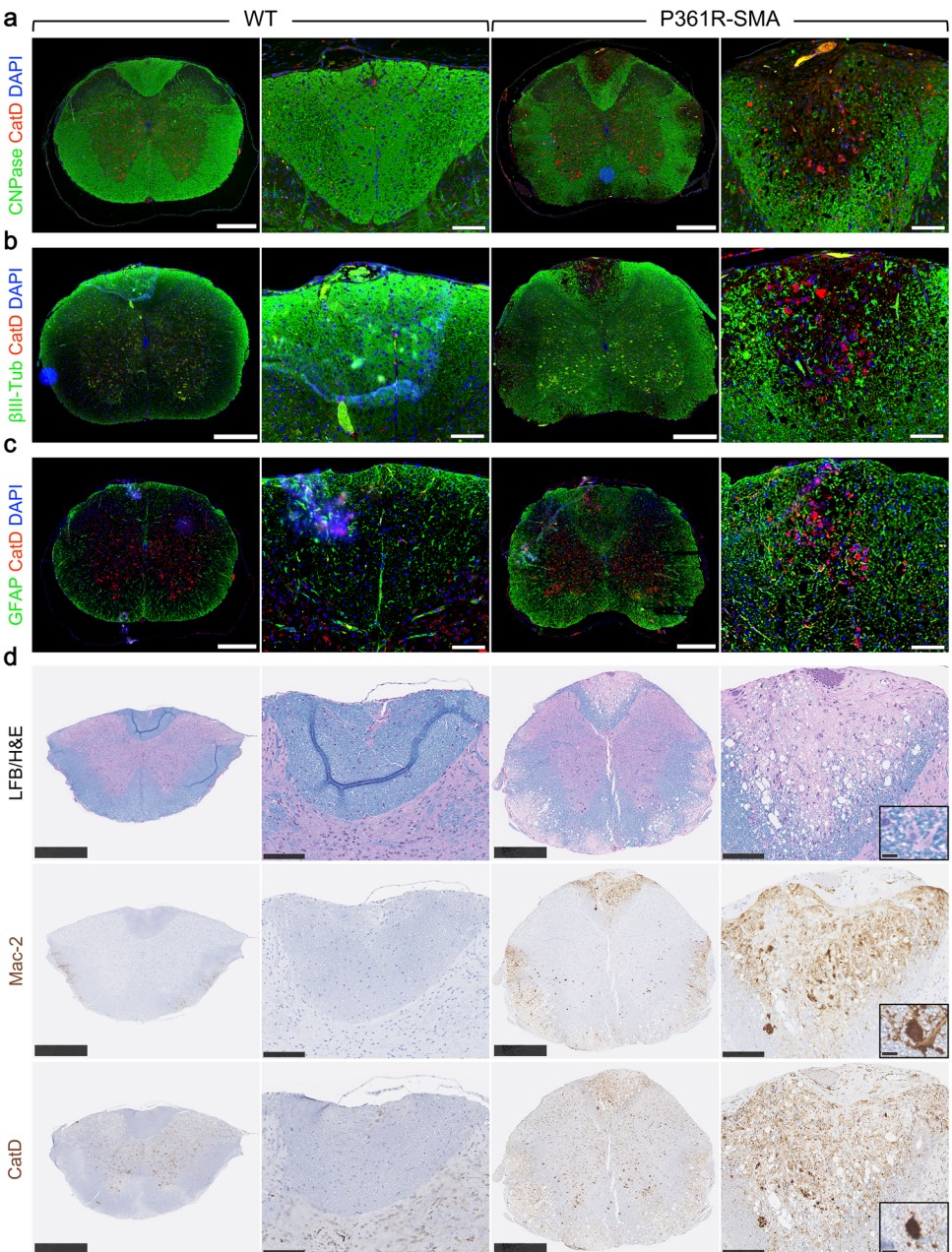

**Fig. 6 Immunofluorescent staining for myelin, neurons and phagocytic cells in spinal cord sections from 21-23-week-old P361R-SMA mice.** Staining for CNPase, a myelin-associated protein (**a**), neuron-specific βIII-Tubulin (βIII-Tub) (**b**) along with a marker of lysosomal lumen, CathepsinD (CatD) in spinal cord sections derived from wild-type (WT) and P361R-SMA mice. **c** Astrocytes (GFAP) and lysosomes (CatD) were immunostained in spinal cord sections from WT and P361R-SMA mice. **d** Serial sections from spinal cords of WT and P361R-SMA mice were stained with luxol fast blue (LFB), or immunohistochemically stained for phagocytic macrophages/microglia (Mac-2) and CatD. The insets highlight a macrophage with LFB staining. Scale bars represent 500 μm (left) and 100 μm (right) for each of WT and P361R-SMA. Higher magnifications are shown focusing on the dorsal funiculus. Scale bar for the insets represents 10 μm.

indicated by yellow staining). Cells with granular Mac-2 staining were also observed (Supplementary Fig. 7d), morphologically resembling Mac-2+ cells found in liver and spleen (Fig. 2a, b). This suggests that damage to the spinal cord and macrophage infiltration in P361R-SMA mice precedes any obvious physical signs of disease.

In samples from 15-week-old mice we confirmed broader loss of myelin/axons (Supplementary Fig. 8a, b), accompanied by high levels of CatD+ staining, but without dramatic unstained vacuolation compared to that found in lesions in samples from 21-week-old animals (Fig. 6a–c). GFAP staining indicated pronounced astrocytosis, and astrocytes overlapped with smaller CatD+ lumens (Supplementary Fig. 8c). Mac-2 staining was also abundant in the lesions (Supplementary Fig. 8d). These data confirm the progressive nature of spinal cord white matter pathology, and suggest that the increase in number of macrophages infiltrating in response to injury is also progressive in our P361R-SMA mice.

**Sphingolipids accumulate in spinal cords from P361R-SMA mice.** Sphingolipid levels in spinal cords have not been previously reported in ACDase-deficient mice. We measured sphingolipids

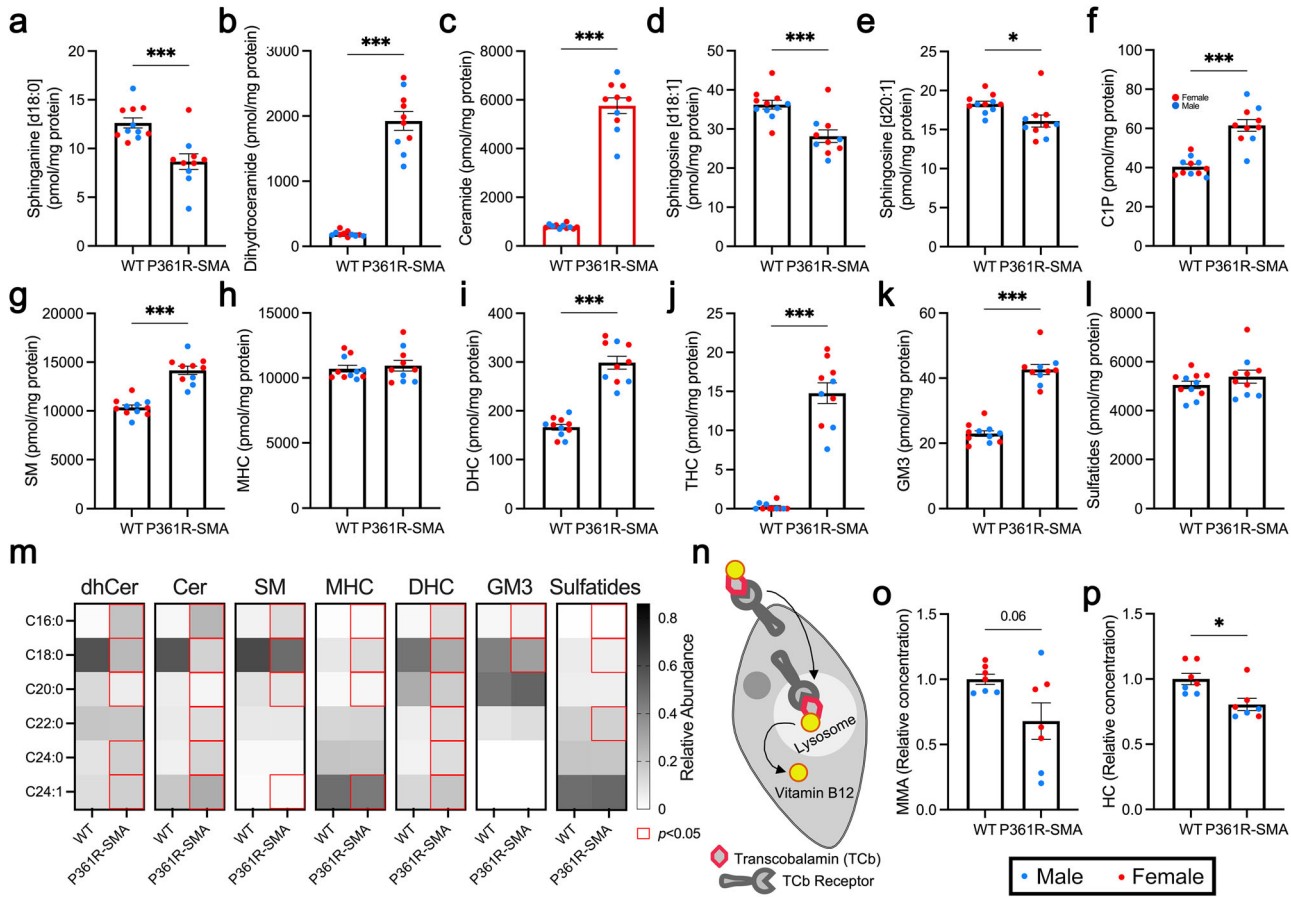

**Fig. 7 Ceramides (Cer) and related sphingolipids in spinal cord homogenates from 21-week-old P361R-SMA mice along with biomarkers of aberrant vitamin B12 metabolism. a–l** Spinal cords from P361R-SMA and wild-type (WT) mice were resected from the cervical to mid-lumbar regions, homogenized, and levels of the indicated sphingolipids were measured by mass spectrometry. Note that sphingosine-1-phosphate was not detected. Molar amounts of individual acyl-chain variants of each species of sphingolipid were summed to determine total levels. Ceramide data is highlighted with red bars. Points represent values from individual mice (WT, $n = 11$ mice; P361R-SMA, $n = 10$ mice), sex is differentiated using different colors (blue for male and red for female), bars represent means and error bars show the standard error of the mean. Data were compared using Welch's $t$ tests, ***$p < 0.001$ and *$p < 0.05$. **m** Relative abundance of the indicated acyl-chain variants for the sphingolipid species measured shown as a heatmap. Data were compared using a two-way analysis of variance with Šidák's multiple comparison test, $p < 0.05$ is indicated with a red outline. **n** Schematic for vitamin B12 (cobalamin) transport into cells via the lysosome. Biomarkers of vitamin B12 deficiency, methylmalonic acid (MMA; (**o**)) and homocysteine (HC; (**p**)), in plasma from WT and P361R-SMA mice. Points represent values from individual mice ($n = 7$) and their color indicates sex (blue for male and red for female). Bars indicate means and error bars show the standard error of the mean. Data were compared using a Welch's $t$ test, *$p < 0.05$. dhCer dihydroceramide, C1P ceramide-1-phosphate, SM sphingomyelin, MHC monohexosylceramide, DHC dihexosylceramide, THC trihexosylceramide, GM3 monosialodihexosylganglioside.

(Fig. 3a) in spinal cord extracts from cervical to approximately mid-lumbar regions. We found a reduction in Spn (Fig. 7a), and large increases in both dhCer and Cer (Fig. 7b, c). We also found significant reductions in both [d18:1] and [d20:1] Sph species (Fig. 7d, e), but were unable to detect S1P. Spn and [d18:1] Sph showed the same trend as the brain. The additional reduction in [d20:1] Sph may be an indication of greater disease burden in the spinal cord, as the longer [d20:1] Sph variant contributes to a large proportion of sphingolipids, for example gangliosides, in the CNS[77,78]. dhCer and Cer also trend in the same direction (Fig. 7b, c), but the changes are much larger than those found in brain lysates (Supplementary Fig. 5b, c). C1P levels are also significantly elevated (Fig. 7f), unlike in the brain (Supplementary Fig. 5g). The relatively dramatic changes in dhCer and Cer, and the increase in C1P, may be further indication of greater disease burden in the spinal cord compared to brain in this model. Comparatively, the levels of these sphingolipids trended in the same direction in P361R-Farber spinal cords

(Supplementary Fig. 9a–f) but increases in dhCer, Cer, and C1P were smaller, suggesting that accumulation of Cer must either reach a threshold to elicit an effect or is not the sole reason for the spinal cord pathology we have observed.

Unlike in the brain (Supplementary Fig. 5h), SM was significantly increased in spinal cords from P361R-SMA mice (Fig. 7g). SM in P361R-Farber spinal cords also trended slightly higher (Supplementary Fig. 9g). MHC, like in the brain, were not significantly changed in spinal cords of either P361R-SMA (Fig. 7h) or P361R-Farber mice (Supplementary Fig. 9h). DHC, on the other hand, were significantly increased in P361R-SMA spinal cords (Fig. 7i) while only a slight difference was measured in P361R-Farber spinal cords (Supplementary Fig. 9i). Total THC and GM3 were also significantly elevated in P361R-SMA spinal cords (Fig. 7j, k); the changes were larger than those found in brains (Supplementary Fig. 5k, l) or in P361R-Farber spinal cords (Supplementary Fig. 9j, k). Sulfatides, like MHC, remained unchanged in spinal cords from P361R-SMA mice (Fig. 7l) and P361R-Farber mice (Supplementary Fig. 9l).

Changes in the abundance of ACVs appear to be much more dramatic in spinal cords of P361R-SMA mice compared to the brain, with perturbations in C16:0 and C18:0 species in particular (Fig. 7m). The relative abundance of C16:0 increased with a concomitant decrease in the abundance of C18:0, except for MHC and sulfatides where both were slightly increased and slightly decreased, respectively (Fig. 7m). Striking perturbations in abundance were seen for dhCer, Cer, and DHC ACVs (Fig. 7m), where the distribution resembles that in spleen (Fig. 3l). This may be a gross indication of the severity of macrophage infiltration into the spinal cord of P361R-SMA mice (Fig. 6d). Additionally, the C16:0 ACVs have the highest fold-increase compared to all other ACVs within each individual species of sphingolipid measured in the spinal cord, and the largest fold-change for each species amongst all organs tested (Supplementary Data 2). Aside from C16:0 species, fold-changes in dhCer, Cer, SM, DHC, THC, and GM3 were greater in longer ACVs, as observed in the brain (Supplementary Data 2). Similarly, while total MHC and sulfatide were not significantly different (Fig. 7h, l), shorter ACVs were significantly increased (Supplementary Data 2). Interestingly, C23:0 and C25:0 MHC were significantly reduced, as in brain, but C23:0 sulfatides were not significantly lower, though they are trending in that direction (Supplementary Data 2). These data suggest that ACDase deficiency leads to complex changes in levels of sphingolipids besides Cer that may be indicators and/or effectors of the profound pathology we observed in P361R-SMA spinal cords.

**White matter loss and neurodegeneration are not caused by vitamin B12-deficiency in P361R-SMA mice.** Our mice present with extensive white matter pathology in the dorsal column, which partially aligns with damage observed in patients with subacute combined degeneration (SACD) of the spinal cord[79,80]. SACD is most often caused by hypocobalaminemia, a deficiency of vitamin B12[81]. Vitamin B12 deficiency can be dietary, a result of malabsorption, or due to defects in cellular processing[82]. Vitamin B12-transcobalamin complexes are taken up by cell surface receptors and trafficked to the lysosome, where the acidic pH and lysosomal proteases allow vitamin B12 to be released (Fig. 7n)[83]. Lysosomal dysfunction has been postulated to affect vitamin B12 processing in some LSDs, causing white matter pathology[84]. Individuals with defects in cellular processing of vitamin B12 are unlikely to present with lower vitamin B12 levels in circulation. Instead, elevated levels of two metabolites, methylmalonic acid (MMA) and homocysteine (HC), can be detected[85]. We measured MMA and HC levels in plasma of P361R-SMA mice and did not find an elevation (Fig. 7o, p), suggesting ACDase-deficiency and/or sphingolipid accumulation do not impinge upon vitamin B12 processing.

**P361R-Farber and P361R-SMA mice have different mutant alleles which interact as compound mutations.** P361R-SMA mice have vastly different pathologies compared to our original P361R-Farber mice. Specifically, dramatic pathology is seen in the liver and spleen of P361R-Farber mice[36,38,44] while the spinal cord is spared, unlike in P361R-SMA mice where the opposite is true. Based on these observations we hypothesized that the presence or absence of the NeoR cassette leads to different mutant alleles capable of interacting with one another as compound mutations and resulting in a mixed phenotype. To test this, we crossed the lines together to obtain mice with these compound mutations in *Asah1* (P361R-NeoR±). P361R-NeoR± mice had significantly shorter survival compared to P361R-SMA mice (median lifespan 127d, $n = 8$, $p < 0.001$ Log-rank test, Supplementary Fig. 10a). Phenotypes were progressive, and included

weight loss, gait abnormality, ataxia, kyphosis, tremors, and incontinence. These signs were very similar to those we observed in P361R-SMA mice and started occurring between ages 11-14 weeks. Surprisingly, post-mortem examination revealed dramatic hepatic, splenic and thymic organomegaly with discoloration and change in turgidity/texture, and enlarged lymph nodes, reminiscent of P361R-Farber mice (Supplementary Fig. 10b). Compound mice also had enlarged bladders, as found in P361R-SMA mice (Supplementary Fig. 10b). Following this cross, we also bred P361R-Farber and P361R-SMA mice with T41A mice. We followed offspring mice for 18–24 months and observed no discernible phenotypes in compound mutants in either cross, or any difference in lifespan compared to littermate controls ($n = 19$–69). This suggests that the T41A mutation is likely inert in mice under normal conditions. The visceral-neuronopathic phenotype combination in P361R-NeoR± mice provides a bridge between the two disease states we have observed in P361R-Farber and P361R-SMA mice. A patient described to present with both Farber-like and SMA-PME-like pathology also suggests that ACDase deficiency is more likely to cause a continuum of disease states and not two delineated and unrelated pathologies[16].

## Discussion

SMA-PME is a devastating ultrarare LSD that has been linked to ACDase deficiency[11]. Patients with SMA-PME, in general, present with progressive lower motor deficits and myoclonic epilepsy, and do not usually have joint contractures and nodules typically found in FD patients[43]. No sex-specific differences have been reported to our knowledge, although clinical data is too sparse to accurately make a conclusive statement. Other visceral pathology, such as histiocytic infiltration, has not been determined for most SMA-PME patients reported. Radiological examination of patients rules out heavy involvement of the brain, but electrophysiology suggests involvement of the CNS[11,15,22,23]. Post-mortem examination of a 69-year-old patient presenting with signs of SMA-PME, showed mild demyelination in the dorsal column, and apparent loss of ventral root cells and degenerative changes in the ventral horn[18]. Loss of normal spinal cord structure after *asah1*-knockdown has been previously reported in zebrafish[11]; the authors showed reduced ACDase activity with developmental defects in motor neuron connections. Ceramide levels, on the other hand, remained unchanged in morpholino-treated zebrafish embryos[11].

To address the lack of a suitable animal model to study pathology, and evaluate treatments for SMA-PME, we first introduced a mutation in ACDase, T41A, in mice. That homologous mutation in humans was found in two siblings, one with a milder form of SMA, and the other not fully described but apparently presenting with more severe disease[14]. However, T41A mice did not present with a different lifespan, or any grossly observable phenotypes over 2 years compared to wild-type animals. They also did not have dramatic changes in sphingolipids in the brain, liver, or spleen. In fact, compound T41A/P361R mutants did not show any phenotypes or changes in lifespan either, strongly suggesting the ACDase T41A mutation is benign in mice.

The discontinuity in spectra of severity observed in different models of ACDase-deficiency published to date prompted us to re-examine our P361R-Farber model. We found that the presence of a NeoR expression cassette resulted in a hypomorphic *Asah1* allele in ear skin fibroblasts. We have not carried out detailed examination of *Asah1* mRNA levels in mice to determine tissue-specific perturbation in *Asah1* expression by the NeoR cassette. However, we observed a profound change in phenotype after removal of NeoR, with animals having a longer lifespan and

reduced visceral pathology. Visceral organs were atrophied in P361R-SMA mice, which we hypothesize is proportional to the loss in body weight observed, as with mouse models of other sphingolipidoses[49,57]. Organomegaly, observed in P361R-Farber mice[36,38], has been postulated to occur due to massive accumulation of foamy macrophages with storage material, which is not seen in P361R-SMA mice. Visceral pathology in P361R-SMA mice was replaced by phenotypic signs and progressive neurological deterioration centered on the spinal cord. P361R-SMA mice developed urinary retention, which is also reported in patients with spinal cord pathologies[86–88]. P361R-SMA mice accumulated ceramide and related sphingolipids, particularly in the spinal cord. As with P361R-Farber mice[36], no evidence was found of any developmental defects in P361R-SMA mice, albeit this was not thoroughly examined. In addition, few differences in the presentation of physiological or biochemical phenotypes were observed between male and female P361R-SMA mice.

A large difference in residual ACDase activity is unlikely to be the sole explanation for variability of phenotypes between T41A, P361R-SMA, and P361R-Farber mice, as we showed dramatic reduction in mouse ACDase activity with T41A and P361R in an overexpression system (Fig. 1c). Note that various mutations may also have different impact on ACDase activity and/or stability in mouse cells compared to human, and in vivo versus in vitro. Deeper investigation with a mouse cell-specific ACDase assay is required to determine the exact cause of differential substrate accumulation and phenotypic variation amongst the various strains presented in this manuscript. Others have proposed that cell-specific levels of ACDase activity and various sphingolipids may be required for function, and that specific mutations may exert a higher burden on specific isoforms of ASAH1[8]. Sphingolipids may also accumulate in various cell types with different kinetics in ACDase deficient mice, such that pathogenic levels are reached in different tissues at different times. It is plausible that T41A does not affect substrate levels in mouse tissue, but that subtle or cell-specific increase in residual ACDase activity in P361R-SMA mice that is uninterrupted by the hypomorphic allele attenuates the visceral FD-like phenotype. In turn, this may allow persistent accumulation of sphingolipids in the spinal cord over time, and the emergence of an SMA-like phenotype. Some examples of cell types of interest are discussed further below.

We found markedly increased Mac-2+ foamy macrophage infiltration into the white pulp of P361R-SMA mice (Fig. 2a), but the germinal center structure was relatively conserved compared to that from P361R-Farber mice[44]. In fact, histological findings in P361R-SMA spleen appeared grossly similar in severity to those found in LV/ASAH1-treated P361R-Farber mice[36]. In liver we found only perivascular infiltration of Mac-2+ histiocytes (Fig. 2b), unlike the wide-spread parenchymal infiltration observed in P361R-Farber mice[38,44]. On the other hand, we found dramatic Mac-2+ staining in spinal cord lesions of P361R-SMA (Fig. 6d) that was not seen in P361R-Farber mice. Mac2+ staining is associated with activated and phagocytosing macrophages/microglia[89,90], including those found in CNS injury and neuroinflammatory diseases[91,92]. Mac-2+ staining was observed in spinal cord lesions starting as early as 8 weeks of age and overlapped with abnormal CatD staining, especially at younger ages, suggesting phagocytic cells contain the bulk of storage material. However, unlike visceral tissues, a large amount of Mac-2+ staining had elongated, spindle-like structure, akin to that of reactive astrocytes. Astrocytes were also elevated in lesions as determined by staining for GFAP (Fig. 6c). Mac-2 also marks astrocytes that have become phagocytic[93]. Small levels of CatD staining were also observed in GFAP+ cells especially in 15-week-old P361R-SMA mice, indicating astrocytes may also contribute to substrate clearance. However, we also found fibrotic scarring in lesions, which has been shown to occur only when immune cells infiltrate in models of demyelinating disease[71,72]. MCP-1, a chemokine that mediates macrophage chemotaxis and infiltration into inflamed tissue[94], was also increased in P361R-SMA plasma. As such, it is highly likely that, in addition to tissue-resident phagocytic cells, bone-marrow-derived macrophages are infiltrating into lesions to clean up damaged myelin and dead cell debris, as shown in mouse models of demyelination[95]. Breakdown of sphingolipid-containing cell debris and myelin by macrophages likely results in additional ceramide formation, which cannot be further catabolized efficiently in ACDase-deficient mice, exacerbating neuropathology. Further work is required to clarify the role of these cells in P361R-SMA spinal cord pathology.

The pathology we have described in the spinal cord of P361R-SMA mice using ex vivo MRI and histological studies appears to be restricted to the white matter—no gray matter findings, for example atrophy or storage pathology in cell bodies, or immune cell infiltration, was found. Physical examination of patients with SMA related to SMN1-deficiency suggests involvement of lower motor neurons, which relate to atrophy of cell bodies in the gray matter of the spinal cord; white matter is usually spared[26]. Anterior horn cell pathology, primarily storage and distention, and loss of neurons is reported in some LSDs[56,96–98]. MRI studies on SMA-PME patients with confirmed mutations in ASAH1 have focused mostly on the brain with mild or no radiological findings[11,15,22,23,99]. To our knowledge, the spinal cord in only one patient with confirmed mutations in ASAH1 has been examined, but the authors only describe their findings as a homogenous loss of spinal cord volume[15].

Our findings in P361R-SMA mice suggest heavy involvement of spinal cord white matter with demyelination, axon degeneration, fibrotic scarring, and disruption and vacuolization of normal structure. These pathological changes were accompanied by infiltration of Mac-2+ macrophage-like phagocytic cells. Such findings are reminiscent of leukodystrophies, a set of inherited demyelinating disorders. Some leukodystrophies are also LSDs with sphingolipid accumulation, i.e., Krabbe disease (OMIM #245200) and metachromatic leukodystrophy (MLD; OMIM #250100). It must be noted that leukodystrophies canonically involve the brain, and clinical pictures usually involve cognitive disability[33,100]. The absence of dramatic imaging findings and other brain involvement excludes a diagnosis of leukodystrophy when considering the clinical history of SMA-PME patients. Further work, especially detailed radiological examination of patient spinal cords by diffusion tensor imaging, for example, is required to clarify whether SMA-PME is better described as a spinal cord-restricted leukodystrophy-like disease or a lower motor neuron disease. MCP-1 levels have also not been evaluated in SMA-PME patients to our knowledge and may provide a simple answer for whether macrophage dysfunction is present, like in our mouse model. Increased MCP-1 levels in SMA-PME patients may also be used to distinguish them from patients with variations of SMA associated with other genetic deficiencies, for example SMN1-deficiency, where MCP-1 levels are inversely correlated with disease severity[101].

Myoclonic seizures are generally caused by perturbed electrical activity in the CNS and have been reported in some LSDs[102]. ACDase deficiency in humans is associated with progressive myoclonic epilepsy[11]. Myoclonic seizures can also occur in patients suffering from diseases with white matter degeneration[103–107]. In addition, numerous deficiencies affecting sphingolipid homeostasis or causing sphingolipid storage (besides sphingolipidoses already classified as leukodystrophies) have white matter abnormalities[30–32,108–111]. To determine whether P361R-SMA mice exhibit increased susceptibility to seizure

development, we used the flurothyl model of repeated generalized seizure induction. Kindling progression induced by flurothyl exposure has been investigated in several inbred mouse strains[112]. It was reported that mice that did not show decreased latency to generalized seizure across training trials had a lower generalized seizure threshold on the first day of flurothyl exposure than in those strains susceptible to kindling. In our study, both P361R-SMA and WT mice show an even lower latency to seizure on Day 1 of training than previously investigated strains[54,112]. It has also been shown that mice that did not develop flurothyl-induced kindling during a training session, did not show an alteration in generalized seizure phenotype upon retesting[112]. Consistent with these data, we did not observe alterations in susceptibility to induced generalized seizures upon retesting 20-week-old P361R-SMA and WT mice. In our study, when 20-week-old P361R-SMA mice were exposed to flurothyl they exhibited an increased frequency of flurothyl-induced myoclonic jerks in comparison with age-matched WT animals. The kindling model may have sensitized mutant mice to develop myoclonic jerks, which are a behavioral correlate of myoclonic seizures. However, the training took place when mice were 15-weeks-old while the retest was done when mice were 20-weeks-old, thus, we cannot discount the effect of increasing age. These data suggest that P361R-SMA mice are prone to the development of myoclonic epilepsy but not generalized seizures. We hypothesize that this may be due to the reduced severity of brain pathology we have observed in our mouse model compared to spinal cord pathology, as generalized tonic-clonic seizures are most often associated with perturbed electrical activity in the brain while myoclonus has been associated with the spinal cord[113,114]. Further studies, including electromyography, are needed to determine overlap of clinical signs of SMA-PME with P361R-SMA mice.

White matter pathology in P361R-SMA mice implies a role for oligodendrocytes. Storage pathology was previously found in oligodendrocytes in the brain of 9-week-old P361R-Farber mice[40]. In addition, others have shown that oligodendrocytes are sensitive to perturbations in sphingolipid levels. For example, a few studies report roles of sphingomyelinases—hydrolases that catabolize sphingomyelin into ceramide—in CNS injury and apoptosis of oligodendrocytes in mouse models of multiple sclerosis[115–118]. Maintenance of sphingolipid homeostasis and de novo Cer synthesis are central to preservation of an intact myelin sheath via oligodendrocyte maturation and function[119]. The central role of ceramide in oligodendrocytes is further exemplified by myelin pathology in patients with ACER3-deficiency[111] and DEGS1-deficiency[108]. Furthermore, Cer, which can also be produced in the CNS following inflammation and injury[120], can activate TLR4[121], which in turn can control oligodendrocyte proliferation during remyelination[122]. A complex circuit of sphingolipid metabolism and signaling appears to play a prominent role during normal and reparative maintenance of myelin. Cer metabolism is impaired in P361R-SMA mice, which may cause broader lysosomal dysfunction and secondary accumulation of metabolites that prospectively impacts sphingolipid-related signaling. Further studies of oligodendrocytes and the pathological role of sphingolipids that accumulate in P361R-SMA mice are warranted.

We did not observe loss of neurons in the motor cortex of P361R-SMA mice. However, we have not thoroughly examined the spinal cord for loss of lower motor neurons. In contrast to white matter pathologies outlined above, a recently described form of childhood amyotrophic lateral sclerosis (ALS) has been associated with dominant gain-of-function mutant alleles of *SPTLC1*[123]. Homeostatic dysregulation of de novo synthesis leads to increased Cer, which may be mediating loss of motor neurons[123]. Accumulation of Cer has also been linked to

pathogenicity in ALS in adult patients and mouse models of the disease[124]. As such, motor neuron involvement cannot be excluded in our mice and in patients with ACDase-deficiency, and Cer accumulation alone may not be the sole cause of the pathology we observe.

We found a dramatic increase in Cer in the spinal cord, and an increase in other tissues as well (Figs. 3d, o, 7c, and Supplementary Data 2). We also found a significant increase in other sphingolipids downstream of Cer, which may indicate broader lysosomal dysfunction and/or disruption of sphingolipid homeostasis. Elevated Cer and substantial secondary elevation of sphingolipids may also relate to the severity of pathology in respective tissues. For example, we found lower Cer, and little or no changes in more complex sphingolipids in the brain compared with the spinal cord, highlighting the spinal cord-centric pathology in P361R-SMA mice. We also found differences in the levels of sphingoid bases when comparing WT and P361R-SMA mice. However, trends in sphingoid base concentrations differed from those we have previously published for P361R-Farber mice[38,40,44], but matched those found in control P361R-Farber animals examined in this study. We hypothesize this may be due to different, tissue-specific changes in expression of ceramidases besides ACDase due to a change in genetic background. However, while every care was taken to ensure reproducible results, we cannot discount technical differences. Nonetheless, the changes are not as dramatic as other sphingolipids, and the effect of relatively small disruptions is unknown. Future studies comparing sphingoid bases and sphingolipids in various tissues between the parent P361R-Farber strain (mixed background) versus the congenic line used in this study may clarify what differences arise from genetic background.

We also found increased dhCer in all tissues we examined. We have previously reported a similar trend in P361R-Farber brains[40]. We have also measured small but significant increases in C12:0 and C24:1 dhCer in FD patient plasma[50]. Since no reported mechanism exists for reduction of Cer to dhCer, de novo synthesis must be maintained or potentially increased. ORMDL proteins on the ER membrane are involved in homeostatic regulation of de novo ceramide synthesis by sensing Cer and, to a lesser extent, dhCer[125,126]. Because ACDase is a lysosomal protein, we expect the excess Cer we measure to be trapped in the lysosome. The increases in CatD-positivity we observed (Fig. 6) supports this hypothesis. As such, the accumulated Cer may not be able to provide feedback to ORMDL proteins, and a specific requirement for extra-lysosomal ceramide may prompt de novo synthesis. However, this does not explain the incomplete conversion of dhCer to Cer. Some unknown feedback mechanism may exist that results in dihydroceramide desaturase inhibition. dhCer may also be bioactive and recycled in the lysosome by ACDase, which would explain the excess accumulation. This hypothesis is supported by in vitro studies showing that human ACDase can catabolize dhCer[127,128]. Determining the subcellular location of dhCer and Cer storage, for example by immunostaining for these lipids[129,130], or by measuring sphingolipids in isolated organelle fractions[131], may help clarify why and where dhCer accumulates in P361R-SMA mice.

We examined individual sphingolipid species and their ACVs to generate hypotheses of specific pathological processes. We found an ~70-fold increase in C16:0 Cer in the spinal cord. Others have shown that an elevation in C16:0 Cer is associated with apoptosis and inflammation[132,133]. Increased C16:0 Cer coupled with an increase in cleaved-Caspase 3 suggests high occurrences of cell death. Further work is required to determine cell types affected but are likely to include oligodendrocytes given the demyelination we find. Conversely, we found no or little change in the absolute level of C18:0 ACV of not only Cer but

also SM, DHC and GM3. The C18:0 ACV of these species is the most abundant in the CNS, specifically in the gray matter[134,135]. The small change in C18:0 species we observe in P361R-SMA may provide further indication of the lack of any gray matter pathology.

Astrocyte activation has been shown to result in increased C16:0 and/or C18:0 DHC synthesis[136]. The elevation in C16:0 DHC in P361R-SMA spinal cords is congruent with the level of astrocytosis we observe and shows promise as a biomarker of reactive gliosis. We also found a relative increase in levels of THC in P361R-SMA spinal cords. However, THC accumulation in brains of mice deficient in α-galactosidase A, which is over 20-fold higher than in P361R-SMA mice, does not result in neuronopathic phenotypes[137,138]. As such, the increase in THC alone is unlikely to contribute to the pathology observed. In contrast, we did not measure significant changes in MHC or sulfatides in CNS tissues (Fig. 7m). Instead, we found both significant increase in shorter ACVs and decrease in longer ACVs (Supplementary Data 2). Differentiating oligodendrocytes express shorter ACVs of sulfatide, and mature, myelinating oligodendrocytes are marked by longer ACVs[139]. The distribution of MHC and sulfatide ACVs in P361R-SMA spinal cords, coupled with the demyelination we observe, suggests oligodendrocytogenesis is occurring to compensate for loss of myelinating oligodendrocytes.

Herein we describe three essentially different models of ACDase-deficiency related to a single mutation, all congenic to the well-studied C57BL/6 J background: P361R-Farber, P361R-SMA and P361R-NeoR±. These models provide a way to study the various types of pathology caused by ACDase-deficiency, they allow the testing of treatments targeting the various affected organs, and they facilitate studies on how to ameliorate combinations of phenotypes.

## Methods

**Mouse lines**. All animal work was conducted per protocols approved by the MCW IACUC. Mice were housed at the MCW Biomedical Resource Center and were provided chlorinated water and a standard autoclaved diet *ad libitum*. C57BL/6N-*Asah1*[T41A] (T41A) mice were generated by The Center for Phenogenomics (Toronto) under contract with our lab. Briefly, Cas9 mRNA and an sgRNA (TTATCTGCGCTCACTCACGT) were injected into C57BL/6 N embryos to introduce a single nucleotide mutation in exon 2, resulting in a single amino acid substitution (p.T41 > A). B6.129-*Asah1*[P361R;Tg(Pgk-NeoR)] (P361R-Farber) mice were generated previously[36]. C57BL/6 J mice for backcrossing, and B6-Tg(Sox2-cre)1Amc mice for germline deletion of the floxed Neomycin-resistance cassette (NeoR) present in the parent P361R-Farber line were purchased from The Jackson Laboratory (Stock no. 000664 and 008454, respectively). T41A mice and P361R-Farber mice were backcrossed to C57BL/6 J mice for 3 and 6 generations, respectively. P361R-Farber mice were crossed to B6-Tg(Sox2-cre)1Amc mice and double heterozygote females were backcrossed to C57BL/6 J mice. Offspring were screened for deletion of NeoR (P361R-SMA) by conventional PCR using Q5 polymerase (New England Biolabs M0493S) followed by Sanger sequencing (Retrogen) of the amplicons using the following primers: 5′-GGTAAGAGAGACAACATAGGTGC-3′ and 5′-AAGGTATGCGGCATCGTCAT-3′. P361R-SMA mice were further backcrossed to C57BL/6 J mice for 7 generations. Genome SNP scanning (The Jackson Laboratory) was used to confirm that all mice were isogenic to C57BL/6 J.

**Breeding and genotyping**. T41A, P361R-Farber, and P361R-SMA mice were maintained as heterozygotes unless indicated otherwise. T41A mice were genotyped using a multiplexed RT-PCR based allelic discrimination assay. Briefly, the following primers were used to amplify the locus with the base substitution: 5′-ACAGCATGCACACAGGATAA-3′ and 5′-TTAGTGGACAGAAGATTGCAGAA-3′. Mutant and WT alleles were distinguished using the following fluorescently labeled locked nucleic acid (LNA) probes: 5′-FAM-CC + A + G + CGT + G + AGT-IB-3′ for the mutant allele and 5′-HEX-TG + G + ACC + A + A + CGT + GA-IB-3′ for WT. gDNA for genotyping was extracted from tail clips using a GeneJET purification kit (Thermo Scientific K0722). Reactions were carried out using PerfeCTa qPCR ToughMix (Quantabio 95114-250). Reaction mixtures contained 500 nM primers along with 250 nM probe, and were conducted over 40 cycles at an annealing temperature of 60 °C on a Quant Studio 5 (Thermo Scientific) instrument. P361R-Farber and P361R-SMA mice were genotyped using the following primers: 5′-TCATAGATGGTAGCAAAGGAGAGATTCT-3′ forward primer, 5′-

AAGGGAGGTGGCTTTGGAAG-3′ specific for the WT allele and 5′-ATTAAGGGTTATTGAATATGATCGGAATTC-3′ specific for the mutant allele. Presence of Cre transgene was tested using the following primers: 5′-GAACCTGATGGACATGTTCAGG-3′ and 5′-AGTGCGTTCGAACGCTAGAGCCTGT-3′. Conventional PCR using alkaline-processed ear tissue as a template and 2x Red-Taq master mix (AS ONE AO180306) over 38 cycles at an annealing temperature of 60 °C run on a Veriti thermocycler (Thermo Scientific) was used to identify Cre-positive mice, or to distinguish mutant and WT amplicons by size on an agarose gel.

**Verification of mutations in *Asah1* transcripts**. Blood was collected from the saphenous veins of T41A, P361R-SMA, and WT mice. RNA was extracted using the TRIzol LS reagent (Invitrogen 10296028) per the manufacturer's instructions. Total cDNA was synthesized from 50 ng of RNA using SuperScript II reverse transcriptase (Invitrogen 18064071) and Oligo(dT)[12-18] primer (Invitrogen 18418012) per the manufacturer's instructions. The coding sequence from *Asah1* transcripts was amplified from total cDNA using Q5 polymerase (NEB M0493S) and the following primers: 5′-CCCAGGGTCGCTGATCAC-3′ and 5′-TTCGGGTCTCAGTACGTCCT-3′. PCR products were extracted using a QIA-quick PCR Purification Kit (Qiagen 28104) and subjected to Sanger sequencing (Retrogen) using the same primers.

**Ear skin fibroblasts**. P361R-Farber mice and WT littermates were euthanized by asphyxiation. Ears were briefly transferred to 70% ethanol, allowed to dry for a few seconds then rinsed in PBS. Ears were macerated and incubated in DMEM with Liberase TM (Roche 05401127001) for 1 h at 37 °C. Digested tissue was washed 2–3 times with PBS then seeded in complete DMEM media supplemented with 1x Pen Strep Glutamine (PSQ, Gibco 10378-016) and 15% v/v heat-inactivated fetal bovine serum (FBS, Gibco 26190-079) for 7-10 days. Cells were passaged every 3–5 days or when 85–90% confluency was achieved. Media was replaced with complete MEM (Thermo Fisher 11095080) supplemented with 15% FBS, PSQ, 1 x non-essential amino acids (Gibco 11140050) and 1 mM sodium pyruvate (Gibco 11360070) to allow selective growth of fibroblasts. Cells were then treated with 20 nM genistein (Sigma-Aldrich G6649) for 3 days to induce overexpression of *Asah1*[140].

**Real-time PCR**. Fibroblasts treated with genistein were harvested in TRIzol reagent (Invitrogen 15596026) and phase separation was then carried out using chloroform per the manufacturer's instructions. RNA was precipitated from the aqueous phase with isopropanol, then transferred to RNeasy columns (Qiagen 74104) for clean-up per the manufacturer's instructions. RNA was eluted in Rnase-free water. RNA concentration was determined using a NanoDrop ONE spectrophotometer (Thermo Scientific). RNA was reverse transcribed using the SuperScript VILO cDNA synthesis kit (Invitrogen 11754-050) per the manufacturer's instructions. A TaqMan gene expression assay (ID: Mm00480021_m1) was used to measure *Asah1* transcript levels in 15 ng of cDNA. Reactions (40 cycles with an annealing temperature of 60 °C) were multiplexed in TaqMan gene expression master mix (Applied Biosystems 4369016) on a Quant Studio 5 (Thermo Scientific) instrument with a *Gapdh* gene expression assay (ID: Mm99999915_g1) to which *Asah1* expression was normalized. Relative quantification was determined using Quant-Studio software (Applied Biosystems v1.3).

**m*Asah1*[P361R] and m*Asah1*[T41A] plasmid constructs and ACDase activity**. Mouse *Asah1* cDNA (NM_019734.3) was synthesized (Genscript) with wild-type sequence or with point mutations leading to p.P361R or p.T41A. Coding sequences were sub-cloned into the pCAGGS expression plasmid, a kind gift from Dr. Razqallah Hakem. Plasmids were propagated in XL10-gold ultracompetent cells (Agilent 200314) and were purified using a miniprep kit (Qiagen 27104). HEK293T cells (CRL-3216, ATCC) were maintained in DMEM medium supplemented with 10% v/v FBS and 1xPSQ. The cells were transfected with plasmids using Lipofectamine 3000 (Thermo Scientific L3000-001) per the manufacturer's instructions and harvested 3 days later by scraping. Cell pellets were resuspended in 0.2 M sucrose with protease inhibitors (Thermo Scientific 78429) and cell lysis was carried out by 5 cycles of freeze-thaw. Lysates were clarified by centrifugation and were stored at −80 °C. ACDase activity was determined using a fluorogenic substrate (RBM14-12)[141]. Cell lysates were mixed with 30 μM RBM14-12 substrate (RUBAM, IQAC-CSIC) in 20 mM sodium acetate (Fisher Scientific, BP334) buffer at a pH of 4.5. The enzyme reaction was carried out in an opaque microtiter plate for 3 h at 37 °C. The reaction was stopped by sequential addition of 50 μL methanol and 100 μL 0.1 M glycine with 2.5 mg/mL sodium periodate (Sigma-Aldrich 311448) at a pH of 10.6, followed by a 2-h incubation at 37 °C. Fluorescence was measured with excitation at 355 nm and emission at 460 nm (Varioskan LUX, Thermo Scientific). The concentration of reaction product was quantified by interpolation from a calibration curve. Specific activity is reported as nmol of product per hour per total protein. Protein concentration was quantified using a bicinchoninic acid (BCA) assay kit (Pierce 23225).

**MCP-1, homocysteine and methylmalonic acid quantification**. Blood was collected from 21–23-week-old mice by cardiac puncture into $K_2$EDTA tubes, and plasma separated by centrifugation ($1000 \times g$ for 5 min at 4 °C) and stored at

−80 °C. MCP-1, homocysteine, and methylmalonic acid levels were determined using commercially available kits (BioLegend 432701, Abcam ab228559, and Aviva Systems Biology OKEH02609, respectively) per the manufacturer's instructions.

**Histology**. Mice were euthanized by $CO_2$ asphyxiation and cardiac puncture; various organs were dissected, rinsed in PBS and fixed in 10% neutral buffered formalin (Sigma-Aldrich HT501128) for 2–3 days. Whole spinal columns were also dissected, cleared of most muscle and tendons, and tied to wooden tongue depressors using cotton thread to prevent curving during fixation. Following fixation, spinal columns were sectioned into four 6–7 mm pieces. Fixed tissue was rinsed and stored in PBS at 4 °C. Spinal columns were additionally decalcified using immunocal reagent (StatLab 1414-32) for 2–3 h and rinsed thoroughly with water. Tissue specimens were processed, embedded in paraffin, and sectioned at 4-8 µm. Sections were stained with hematoxylin and eosin (H&E). Brain and spinal column sections were stained for myelin with Luxol Fast Blue (LFB) and counter-stained with H&E[142]. Spinal cord sections were also stained using Masson's trichrome, and a modified Bielchowsky silver stain with periodic acid-Schiff stain[142]. Sections for visceral organs and spinal cords were also evaluated by immunohistochemistry (IHC)[38,41] using the following primary antibodies: anti-Mac-2 clone M3/38 (1:600) (Cedarlane CL8942AP), anti-Lamp1 clone D2D11 (1:400) (Cell Signaling 9091 S) and anti-Cathepsin D clone EPR3057Y (1:1000) (Abcam ab75852). Processing, staining and IHC was performed by the Children's Research Institute Histology Core (Milwaukee, WI) as fee-for-service. Slides were scanned using a NanoZoomer 2.0-HT histology slide scanner and images viewed on NDP.view2 software (Hamamatsu Photonics).

**Western blotting**. Livers and spinal cords were homogenized in water containing Protease and Phosphatase Inhibitor Cocktails (Thermo Scientific 78429 and 815-968-0747, respectively) using a Bullet Blender Storm 24 tissue homogenizer (Next Advance) with 0.5 mm zirconium oxide beads. Samples were diluted immediately in RIPA buffer (50 mM Tris, 150 mM NaCl, 1% Triton X-100, 1% sodium deoxycholate, 0.1% SDS, 1 mM EDTA) and further homogenized, then clarified by centrifugation (19000 × $g$ for 10 min at 4 °C). Clarified lysates were stored at −80 °C. Protein concentration was determined using a bicinchoninic acid (BCA) assay kit (Pierce 23225). Equal amount of protein (15 µg) for each sample was separated by SDS-PAGE and subsequently transferred to a PVDF membrane. Membranes were blocked with 5% milk and probed using primary antibodies against the following proteins: Cathepsin D (1:5000) (ab75852, Abcam), Caspase-3 (1:10000) (9664 S), STAT3 (1:1000) (4904), p-STAT3 (1:1000) (9145), NF-κB (1:1000) (8242), p-NF-κB (1:750) (3033), MBP (1:1000) (78896 S), MOG (1:1000) (96457 S) and MAG (1:1000) (9043 T, all from Cell Signaling Technologies). -Actin (1:10000) (a3854, Sigma-Aldrich) levels were measured as a loading control. HRP-conjugated anti-Rabbit IgG (1:10000) (A6154, Sigma–Aldrich) was used as secondary antibody. Immunoreactivity was detected using chemiluminescent HRP substrate and blots were imaged on a ChemiDoc MP imager. Densitometry using ImageJ (v1.51 NIH)[143] was used to quantify signals in relevant bands. Band intensities for relevant proteins were normalized to the signal from β-Actin, then to the average signal from WT mice, and are displayed as relative changes from WT.

**Quantification of sphingolipids**. Mice were euthanized by $CO_2$ asphyxiation and visceral organs and the brain were removed, rinsed thoroughly in PBS, and snap frozen on dry ice. Fat and muscle were resected away from the spine, vertebrae were laterally broken using sharp scissors, and the tissue lifted off to expose the spinal cord. The spinal cord was resected using thin forceps and snap frozen on dry ice. Tissues were thawed on ice and homogenized by sonication (Misonix, Farmingdale, NY) in 0.02 M Tris (containing 0.5 M NaCl and 0.1% v/v Nonidet P-40; pH 7.0). Total protein was estimated by the method of Lowry et al. (1951)[144]. In brief, 100 µL of diluted homogenate was mixed with 1 mL reagent D (392 µM copper sulfate and 569 µM sodium citrate with 0.2 M sodium carbonate and 0.1 M sodium hydroxide). Folin-Ciocoulteau's reagent was added (0.1 mL; 1:1 dilution with water) and the homogenates vortexed. Samples were incubated at RT for 30 min. Absorbance was measured on a Victor Nivo Multimode Microplate Reader (PerkinElmer, Waltham, MA) at wavelength 750 nm; a standard curve (0-320 µg/mL protein) was used to calculate the protein concentrations.

With the exception of the phosphorylated species, all sphingolipids were extracted from 50 µg of protein in 10 µL of homogenate using a single-phase lipid extraction by the addition of 0.2 mL chloroform/methanol (2/1) and 5 µL of internal standard (IS) mixture containing 10pmol each of Cer(d18:1/17:0) [N-heptadecanoyl-D-erythro-sphingosine], MHC(d18:1/15:0) [N-pentadecanoyl-psychosine], DHC(d18:1/16:0-d$_3$) [N-palmitoyl-d3-lactosylceramide], THC(d18:1/17:0) [N-heptadecanoyl ceramide trihexoside], C12 sulfatide [3-O-sulfo-D-galactosyl-β1-1′-N-lauroyl-D-erythro-sphingosine], and 5pmol each of sph d17:1 [D-erythro-sphingosine (C17 base)], spn d17:0 [D-erythro-sphinganine (C17 base)], dhCer(d18:0 /8:0) [N-octanoyl-D-erythro-sphinganine], and SM(d18:1/12:0) [N-lauroyl-D-erythro-sphingosylphosphorylcholine]. All standards were from Avanti Polar Lipids (Alabaster, AL) with the exception of MHC, DHC and THC, which were purchased from Matreya LLC (State College, PA). Samples were mixed for 10 min, sonicated for 30 min, and allowed to stand at RT for 20 min. Samples were then centrifuged (13000 × $g$; 10 min at RT) and the supernatant was

removed and dried down under $N_2$ at 40 °C[145,146]. Lipid extracts were reconstituted in 10 mM ammonium formate in methanol, and partial separation was achieved on an Agilent Zorbax Eclipse C18 column (2.1 × 50 mm, 1.8 µm) maintained at 40 °C with a flow rate of 0.4 mL/min and a 1 µL injection. Mobile phase A consisted of water/acetonitrile (60/40) with 10 mM ammonium formate, while mobile phase B was 2-propanol/acetonitrile (90/10) with 10 mM ammonium formate. The column was equilibrated at 10% mobile phase B before a linear ramp to 50% at 2 min and a further increase to 100% mobile phase B at 8 min. This was held for 0.5 min before a return to 10% mobile phase B at 9 min. The column was equilibrated for 1 min prior to the next injection. The first 1 min of column flow was diverted to waste before being directed into the electrospray source (ES 5500 V) of a SCIEX QTRAP 6500 triple quadrupole tandem mass spectrometer in positive ion mode. Source conditions included an ion source temperature of 250 °C; curtain gas, 25 units; collision gas, medium; nebulizer gas 1, 20 units and auxiliary gas 2, 40 units. Individual species were quantified by multiple reaction monitoring (MRM) with concentrations calculated by comparing the area of the analyte to that of the corresponding IS[146].

Sphingosine-1-phosphate (S1P) and ceramide-1-phosphates (C1P) were extracted from tissue homogenates with 100 µg of protein as above with addition of 10pmol each of N-lauroyl-ceramide-1-phosphate (C12:0 C1P) and D-erythro-[d17:1]sphingosine-1-phosphate (S1P[d17:1]) as IS (Avanti Polar Lipids, Alabaster, AL). Following reconstitution of the lipid extracts in 0.1 mL of 10 mM ammonium formate in methanol, analytes were partially separated as above with the exception that 0.5% formic acid was added to both mobile phases and 2 µL of extract was injected. The first 1 min of column flow was diverted to waste before being directed into the electrospray source (ES 5500 V) of the mass spectrometer with an ion source temperature of 320 °C. Curtain gas, 25 units; collision gas, medium; nebulizer gas 1, 30 units and auxiliary gas 2, 40 units. Individual species of C1P and S1P were quantified by MRM (Supplementary Table 1) with concentrations determined by relating the peak area of the analyte to that of the internal standard using MultiQuant 3.0.1 (SCIEX, Framingham, MA, USA).

**Behavioral studies**. Muscle strength and motor coordination was assessed by a wire-hang test and an accelerating rotating rod (rotarod) test, respectively[147,148]. For the wire-hang test, mice were placed on a metal mesh surface that was boxed-in by smooth metal walls. The surface was slowly turned upside-down, allowing mice to hold on. The time to fall, reported as hang time, was recorded to a maximum of 2 min. For rotarod testing, mice were placed on a rotarod device (IITC lifescience), which was then accelerated from 4 to 40 rpm over 5 min. The time it took for mice to fall off the rotating rod, reported as latency, was automatically recorded. For both behavioral analyses, mice were tested three times per day for two consecutive days at each reported age with a 10-minute rest between the trials. The mean of the three trials from the second day was used for analyses.

Mechanical sensitivity was tested using a set of 8 manual von Frey monofilaments (Touch Test sensory evaluators, North Coast Medical; 0.07-4.0 g). Percent withdrawal and withdrawal force thresholds were adapted from previously described methods[149,150]. All tests were performed at a fixed time to minimize variability due to circadian rhythm. Mice were placed in plexiglass compartments with dark walls atop a wire floor grid. Before testing, subjects were left undisturbed for 30 min and acclimatized to the tester for a further 30-45 minutes. To determine percent withdrawal, only the 1.0 g filament was applied to the middle plantar region of a hind paw. A positive withdrawal response was recorded when licking, flicking, or swift withdrawal was observed. This was repeated 10 times for each hind paw and the percentage withdrawal averaged for both paws. To determine withdrawal force threshold, all 8 filaments were used. After full acclimatization, the first 0.4 g filament was applied to the middle plantar region of a hind paw. Paw withdrawal, flicking, or licking was marked as a positive response. Depending on whether the response was negative or positive, the next larger or smaller filament force was tested, respectively. Testing continued until two positive responses were recorded for a specific force for each hind paw. The minimum force threshold needed to induce paw withdrawal was expressed as the average of the force resulting in two positive responses across left and right hind paws in each subject.

**Flurothyl-induced seizure susceptibility testing**. Seizure susceptibility of P361R-SMA mice was tested in accordance with a previously published protocol[54]. Briefly, individual mice (starting from 15 weeks of age) were exposed to 10% flurothyl (Sigma-Aldrich, 287571) in closed glass chamber via a syringe pump (Kent Scientific, model: GENIE Plus) at a flow rate of 6 mL/h onto a gauze pad. Once the animal expressed a generalized tonic-clonic seizure, the chamber was opened to fresh air, resulting in rapid elimination of the flurothyl. Mice were given one flurothyl-induced seizure per day for 8 days (the training/induction phase). The time interval between inductions was kept as close to 24 h as possible (± 1 h). Following a 4-week rest period (the incubation period), mice were reexposed to flurothyl as above (rechallenge) and seizure activity recorded. Myoclonic jerks are defined as brief but intense contractions of the neck and whole body musculature that typically precede generalized seizures. Generalized seizures are characterized by generalized clonus of the entire body and repeated clonic jerks of the forelimbs and hindlimbs along with tonic muscle extensions of the limb and trunk muscles. The latency to the first myoclonic jerk (threshold), the frequency of myoclonic

jerks, the latency to generalized seizure, and the duration of the seizure were recorded at the rechallenge phase of the model.

**Brain and spinal cord immunofluorescence**. Mice were euthanized by $CO_2$ asphyxiation and perfused with cold PBS followed by 4% paraformaldehyde (PFA) made in PBS. Brains were resected and left overnight in 4% PFA, rinsed in PBS then left in 30% w/v sucrose solution for 2 days. Brains were then embedded in Cryomatrix resin (Thermo Scientific 6769006), flash frozen in liquid Nitrogen and stored at −80 °C. Sagittal sections were blocked with 5% Bovine Serum Albumin (BSA) and 0.3% Triton X-100 in PBS, then incubated overnight at 4 °C with primary antibodies in 1% BSA and 0.3% Triton X-100 in PBS. The following antibodies were used: mouse anti-mouse neuronal nuclear antigen (NeuN) (1:200) (E4M5P; Cell Signaling Technology), rabbit polyclonal anti-mouse/rat CD68 (1:200) (AB125212; Abcam), rabbit anti-mouse/rat NeuN (1:200) (MABN140; Millipore), and mouse anti-mouse/rat GFAP (1:100) (8-1E7; DSHB). The slides were further stained with Alexa Fluor 555 or 488–conjugated goat anti-mouse IgG (1:2000) and Alexa Fluor 555 or 488–conjugated goat anti-rabbit IgG (1:2000), as appropriate (all from Thermo Fisher Scientific). The slides were also treated with TrueBlack (Biotium) before being mounted with mounting reagent containing DAPI and analyzed using a laser scanning confocal microscope (Leica TCS SP8). ImageJ (v1.51 NIH)[143] was used to quantify signal from appropriate channels.

Formalin-fixed, paraffin embedded sections of spinal columns, as made above, were used for immunofluorescence (IF) studies. Briefly, sections were deparaffinized in xylene, and slowly rehydrated in decreasing concentrations of ethanol. Antigen retrieval was carried out by heating sections at 95 °C in citrate buffer at pH 6.0 (Sigma-Aldrich C9999). Sections were then treated with rodent block M (Biocare Medical RBM961) for mouse-on-mouse staining followed by blocking solution (5% BSA in PBS). Sections were next incubated overnight with primary antibodies against the following: Cathepsin D (1:1000) (clone EPR3057Y, Abcam ab75852), CNPase (1:400) (clone CL2887, Novus Biological NBP2-46617), βIII-Tubulin (1:300) (clone TuJ-1, R&D Systems MAB1195SP), and GFAP (1:300) (clone GA5, Cell Signaling 3670 T). Sections for IF were washed with PBS, then incubated with secondary antibodies conjugated to Alexa Fluor 488 (1:750) or Alexa Fluor 594 (1:1000) (Invitrogen A28175 and A-11012). Washed sections were then mounted in Vectashield mounting medium containing DAPI (Vector laboratories H-1200-10). Whole slides were scanned at high resolution on a VS120 slide scanner (Olympus) using a Plan-APO 20x lens (located at CRI Imaging core, MCW. Images were viewed, and background and brightness adjusted using OlyVIA software v3.2.1 (Olympus). ROIs were exported in the TIFF format, and images were cropped and post-processed using paint.net v4.2.15 (dotPDN, LLC).

**Ex vivo magnetic resonance imaging**. Whole spinal columns that were fixed and sectioned as described above were placed in 0.6 mL microcentrifuge tubes and immersed in the fluorinated and non-protonated oil Fomblin Y (Avg MW 1800, Sigma-Aldrich 317926) to minimize water signals outside of the cord and to reduce susceptibility artifacts. Tubes were sealed with parafilm to ensure no air was present. Tubes were placed in a custom 8 mm × 8 mm solenoid coil designed to fit small samples for enhanced signal-to-noise ratio. Spinal columns were imaged using T2-weighted MRI (Bruker 9.4 T). A fast spin echo sequence was used for three-dimensional imaging at an isotropic resolution of 55 μm$^3$ (192 × 96 × 96 matrix). Two echo times (23.5 and 86 ms) were acquired with a 4000 ms repetition time, turbo factor of 8, and 4 repetitions for a total imaging time of about 5.5 h per sample. Data were converted to NIfTI-1 data format using MRIcroGL software (v1.2.20191219, University of South Carolina). Images from separate repetitions were spatially aligned to account for any scanner drift or movement of the sample during the scan using advanced normalization tools (ANTs)[151]. The eight images from the two echoes and four repetitions were averaged to increase signal-to-noise. Averaged data were visualized, resliced/reoriented, and contrast/brightness was adjusted using ImageJ (v1.51 NIH)[143].

**Statistics and reproducibility**. GraphPad Prism (v9.1.0 for MacOS, GraphPad Software, LLC) was used to perform statistical tests and to plot data. Most experiments were conducted with at least 3 replicates. Statistical tests used for each experiment to report significance are indicated in figure legends, and $p$ values are reported using asterisks in the style of the American Psychological Association (ns $p > 0.05$, $*p < 0.05$, $**p < 0.01$, and $***p < 0.001$). Figure panels containing images are representative of at least 2 mice per group. A table summarizing statistical comparisons by sex is provided in Supplementary Data 3. Raw data used for statistical analyses throughout is attached as Supplementary Data 1. Results of the statistical analyses performed throughout are attached as Supplementary Data 4.

**Data presentation**. Drawings and image collages were created in Microsoft PowerPoint for Microsoft 365 (v16.0 for Windows or v16.52 for Mac). Panels with charts were created in GraphPad Prism (v9.1.0 for MacOS, GraphPad Software, LLC).

**Reporting summary**. Further information on research design is available in the Nature Portfolio Reporting Summary linked to this article.

## Data availability

This study includes no data deposited in external repositories. Raw data for all experiments is available in Supplementary Data 1. Unedited uncropped western blots for Figs. 2 and 5 are presented as Supplementary Fig. 11.

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

## Acknowledgements

We thank Dr. Shaalee Dworski, Dr. Everett Tate, and The Center for Phenogenomics (Toronto, ON, Canada) for assistance with generating and maintaining the T41A mice. We would like to acknowledge technical support from Christine Duris and the Children's Research Institute Histology Core, Dr. Suresh Kumar and the Children's Research Institute Imaging Core, and Matthew Runquist and the Center for Imaging Research for MRI. We would also like to thank Dr. Aron Geurts for assistance with sequencing and Dr. Jakub Sikora for microscopical evaluations. The authors thank Dr. Thierry Levade for critical reading of the manuscript, and Dr. Mary Faber for assistance with revisions. This work was supported by the Midwest Athletes against Childhood Cancer (MACC) Fund Professorship to JAM. Support was provided by the Research and Education Component of the Advancing a Healthier Wisconsin Endowment at the Medical College of Wisconsin. MRI studies were partly supported by funds from the Daniel M. Soref Charitable Trust awarded to JAM and MDB. Immunofluorescence brain imaging was partially supported by an operating grant PJT-156345 from the Canadian Institutes of Health Research to AVP.

## Author contributions

M.S.N. conceived the study. M.S.N., J.R., and J.A.M. planned experiments. M.S.N., J.R., A.K., C.J.A., and M.B. conducted experiments, collected, and analyzed data. J.T.S. and M.F. measured concentrations of sphingolipids. T.M.X. and A.V.P. evaluated brains by immunofluorescence. M.D.B. optimized ex vivo MRI and helped analyze the data. M.W.L. provided histopathological evaluations. O.I. analyzed data from the seizure susceptibility test. M.S.N., J.R. and J.A.M. wrote the original manuscript. MS.N., J.R., W.M.M. and J.A.M. revised the manuscript.

## Competing interests

The authors declare no competing interests.
