## [Peer Review File · Communications Biology]

Reviewers' comments:

Reviewer #1 (Remarks to the Author):

In this study by Nagree et al, they describe a novel mouse model that recapitulates a clinical manifestation that results from a mutation in *ASAH1* that encodes ACDase that is critical for sphingolipid metabolism. Mutations in this gene are associated with rare lysosomal disorders manifesting in two distinct clinical pathologies: Farber Disease (FD) and spinal muscular atrophy with progressive myoclonic and absence epilepsy (SMA-PME). This group has generated mice with P361R mutation in *ASAH1* and have previously characterized recapitulation of many features of FD. In this current study, the group reports a previously undescribed features of the SMA-PME involving the peripheral nervous system and demyelination by removing the NeoR from the original genetic construct. This appears to attenuate the severe phenotype of the P361R-FD mice. The authors posit that this mouse with these distinct features would serve as an important model for the second set of clinical pathologies that are associated with the mutation. Overall, the authors characterizes in detail the overall phenotype of the mouse pathology with in-depth biochemical measurements of all the species of the sphingolipids in different tissues including the brain and the spinal cord. The study is carefully done and the manuscript is nicely prepared, providing a novel model of a rare genetic condition in humans. Some minor concerns are detailed as follows.

What is the significance of organ atrophy in P361R-SMA mice? Is this due to fibrosis from generalized inflammation?

Figure 1H needs better labeling and description. The images are not at all clear based on how they are laid out.

In Fig 2, the representative histological images for spleen would be better presented by flow cytometry.

When comparing the degree of sphingolipid accumulation and comparing between P361R-SMA and P361R-FD mice, it would be informative to give quantitative measures rather than subjective terms used 'less increased...' – line 216-7

Histological assessment of muscle and fat are not conventional and difficult to compare between genotypes (Fig 1H and supplemental Fig 3) – need to specify from which muscle group and which fat depot with similar orientation for objective comparison. Their statement 'Muscle atrophy in P361R-SMA mice was less severe than fat tissue but evident, with a slight perivascular increase in connective tissue staining' line 241 – this description is not scientific and not meaningful. Would need some more objective measures when discussing atrophy of muscle. Comparing between the degree of atrophy between muscle and fat doesn't not have any particular significance and would be best to compare each tissue between genotypes.

Reviewer #2 (Remarks to the Author):

In this manuscript, Nagree et al., develop and characterize a novel mouse model of spinal muscular atrophy-progressive myoclonic epilepsy syndrome (SMA-PME). This work draws upon a previously established model of a related syndrome, Farber disease (FD), whereby a patient mutation (P361R) in the ASAH1 gene was introduced into murine Asah1 via homologous recombination. This recombination event disrupted intronic DNA and here the authors determine that this disruption led to dramatic suppression of Asah1 mRNA along with disruption of ACDase activity by the P361R mutation, leading to a severe FD phenotype. When the authors repaired the disrupted intronic DNA, a model emerged with a less severe phenotype that better reflects the pathological symptoms of SMA-PME. The authors present an impressive and thorough pathological and lipidomic analysis of this mouse model which may be useful for future therapeutic development. It is difficult to develop a perfect model of any disease especially for those that are rare and present with a wide range of clinical symptoms. Thus, this work will be of great interest to a broad field of rare diseases. The reviewer appreciates the progress and novelty in this work, but the following issues should be addressed:

Major concerns

- No experimental evidence of myoclonic seizures is presented. In the discussion, the authors mention that they were "unable to determine whether P361R-SMA mice had any seizure activity." Why is this the case? As PME is an important clinical phenotype of this disease, it seems crucial to determine if these mice have myoclonic epilepsy phenotypes.
- The authors first make a T41A mouse based on another patient mutation. However, these mice are largely asymptomatic. The ACDase activity for this mutation needs to be evaluated (similar to Fig 1D,C). If this mutation also disrupts ACDase activity, it calls into question the conclusion that the lack ACDase activity is etiology behind SMA-PME.
- The histology images compare WT to P361R-SMA mice while the text continues to make comparisons to P361R-Farber mice from previous publications. Representative images of P361R-R mice need to be shown in the same panel as P361R-SMA mice so the reader can make their own comparisons without having to find previous references.
- The lipid analyses in the current form are not extremely valuable to the reader and are somewhat a "data dump." Discussion of lipid changes are largely presented as "lipid X changes in brain region Y" with minimal discussion of the biological relevance for these changes, especially ACVs. This reviewer suggests moving most of the lipid data to the supplement and highlighting only a handful of the most important lipid changes.
- In figure 3, ceramide is increased in both spleen and liver in the P361R-SMA mouse, as one would expect with disrupted ACDase activity. However, it is unclear why the product of this activity is also upregulated. Please explain.

Minor

- In figure 3, "Spleen" overlaps with other text on the left side of the figure (near B)
- Need to label the left side of the bottom panel of figures in Fig 4E.

Reviewer #3 (Remarks to the Author):

In this manuscript entitled "Unique Spinal Muscular Atrophy-Like Phenotype in a Novel Mouse Model of Acid Ceramidase Deficiency", the authors show that a mouse model with spinal muscular atrophy with progressive myoclonic epilepsy (SMA-PME)- like phenotype (P361R-SMA). Although most of the data in this manuscript are convincing and presented by well-designed experiment, I have listed my concerns below:

Major comments

1. Is there any effect of sex difference on the pathological condition? In Figure 1, only weight is shown as males and females, but what about other conditions? Please describe the sex used in the experiment. If you're using both genders, it might be a good idea to color-code each data plot sex. Also, I recommend describing the effects of gender differences in this mouse model and sex differences in general epidemiology in the discussion section.

Minor comments

1. In Figure 1H, It is difficult to understand which organ it is, please add explanations. In the Figure legend, the authors explain that "Mice and organs were placed next to a ruler for accurately scaled resizing of images, such that WT and P361R-SMA images can be directly compared", but it is more accurate to insert the scale bar.
2. In Lines 184-191, Although there is a description to compare the pathology of P361R-SMA mice to P361R-Farber mice, the actual figure is a comparison of Control and P361R-SMA mice. Although this may be the case when compared with previous studies, It may not be appropriate to compare P361R-SMA mice and P361R-Farber mice as result. As shown in the data & figure presented, it would be better to describe that MAC-2 has increased compared to control, and compare P361R-Farber mice of past results in the discussion sections. If it is necessary to compare the pathology of P361R-SMA and P361R-Farber mice, please present the data and perform statistical comparisons.
3. In Figure 2D, you may need to present an image of the Western blot. Also, please show the data of each sample with dot plot like other data.
4. In the lines 394-395 and Figure 6D, although CatD+ cluster is overlapped with Mac-2 in the serial section, the parts that do not overlap are more visible. It would be better to quantify the percentage of overlapping cells.

Response to reviewers

Please see responses in blue text.

Referee expertise:

Referee #1: Mouse models of vascular and metabolic disorders

Referee #2: Metabolic disorders in epilepsy

Referee #3: Mouse models of metabolic and neurological disorders

Reviewers' comments:

Reviewer #1 (Remarks to the Author):

In this study by Nagree et al, they describe a novel mouse model that recapitulates a clinical manifestation that results from a mutation in *ASAH1* that encodes ACDase that is critical for sphingolipid metabolism. Mutations in this gene are associated with rare lysosomal disorders manifesting in two distinct clinical pathologies: Farber Disease (FD) and spinal muscular atrophy with progressive myoclonic and absence epilepsy (SMA-PME). This group has generated mice with P361R mutation in *ASAH1* and have previously characterized recapitulation of many features of FD. In this current study, the group reports a previously undescribed features of the SMA-PME involving the peripheral nervous system and demyelination by removing the NeoR from the original genetic construct. This appears to attenuate the severe phenotype of the P361R-FD mice. The authors posit that this mouse with these distinct features would serve as an important model for the second set of clinical pathologies that are associated with the mutation. Overall, the authors characterizes in detail the overall phenotype of the mouse pathology with in-depth biochemical measurements of all the species of the sphingolipids in different tissues including the brain and the spinal cord. The study is carefully done and the manuscript is nicely prepared, providing a novel model of a rare genetic condition in humans. Some minor concerns are detailed as follows.

What is the significance of organ atrophy in P361R-SMA mice? Is this due to fibrosis from generalized inflammation?

We thank the Reviewer for their insightful question. We have not investigated the exact cause of organ atrophy in our P361R-SMA mice. In P361R-FD mice we have postulated that organomegaly, where observed (e.g. spleen, thymus, liver), is due to accumulation of foamy macrophages with storage material (see Alayoubi et al. 2013 and Yu et al. 2019). We see reduced accumulation of such macrophages in the same organs in P361R-SMA mice. We do not have any evidence of fibrotic damage in the aforementioned tissues. Instead, we postulate that the atrophy is proportional to reduced size/weight of the mice, which may be occurring to compensate for aberrant metabolism in the mice. Generalized atrophy of mouse models of LSDs

is a well-noted phenomenon (see models of Krabbe disease, Niemann-Pick disease, neuronopathic Gaucher disease).

Figure 1H needs better labeling and description. The images are not at all clear based on how they are laid out.

We thank the Reviewer for their feedback and have labelled the various organs in what is now Fig 1H and added a scale bar, based on additional feedback from Reviewer 3.

In Fig 2, the representative histological images for spleen would be better presented by flow cytometry.

We thank the Reviewer for their suggestion and agree that evaluation by flow cytometry would be a better measurement of the finer properties of the various cell types present in the spleen, and comparison of their proportions. However, for the purposes of this study and for comparison to previous findings in the related P361R-FD mouse, we find it important to demonstrate the location of cells with storage material, and how they impact the architecture of the various organs we evaluated, including the spleen. In addition, we have not been able to fully optimize a flow cytometry protocol that allows us to preserve extremely large, foamy cells without lysis, although we note that in previous studies, we still found meaningful differences between WT and P361R-FD mice. Nonetheless, we believe that histological evaluation of the spleen is sufficient to draw the conclusions we have outlined (disrupted structure and accumulation of foamy macrophages) with regards to our mouse models. Future studies outside the scope of this submission should evaluate all organs of relevance to Farber disease and SMA-PME by multiparametric flow cytometry over the lifespan of these mouse models to investigate disease progression at a cellular level.

When comparing the degree of sphingolipid accumulation and comparing between P361R-SMA and P361R-FD mice, it would be informative to give quantitative measures rather than subjective terms used ‘less increased...’ – line 216-7

We thank the Reviewer for indicating the need for better description of our data. We have made the following changes to the text:

“Elevations were more dramatic in the spleen (~30-fold) than in the liver (~4-fold), but overall were less increased compared to previously reported changes in tissues from the P361R-Farber mice (>75-fold in spleen and ~20-fold in liver)^{36,38,43}.”

Histological assessment of muscle and fat are not conventional and difficult to compare between genotypes (Fig 1H and supplemental Fig 3) – need to specify from which muscle group and which fat depot with similar orientation for objective comparison. Their statement ‘Muscle atrophy in P361R-SMA mice was less severe than fat tissue but evident, with a slight perivascular increase in connective tissue staining’ line 241 – this description is not scientific and not meaningful. Would need some more objective measures when discussing atrophy of muscle. Comparing between the degree of atrophy between muscle and fat doesn’t not have any particular significance and would be best to compare each tissue between genotypes.

We thank the Reviewer for their feedback and their suggestion. We have decided to remove this part (the supplementary figure and text noted by the Reviewer) from our manuscript as we agree that atrophy of muscle and fat in our model need to be analyzed more extensively in the future.

Reviewer #2 (Remarks to the Author):

In this manuscript, Nagree et al., develop and characterize a novel mouse model of spinal muscular atrophy-progressive myoclonic epilepsy syndrome (SMA-PME). This work draws upon a previously established model of a related syndrome, Farber disease (FD), whereby a patient mutation (P361R) in the *ASAHI1* gene was introduced into murine *Asah1* via homologous recombination. This recombination event disrupted intronic DNA and here the authors determine that this disruption led to dramatic suppression of *Asah1* mRNA along with disruption of ACDase activity by the P361R mutation, leading to a severe FD phenotype. When the authors repaired the disrupted intronic DNA, a model emerged with a less severe phenotype that better reflects the pathological symptoms of SMA-PME. The authors present an impressive and thorough pathological and lipidomic analysis of this mouse model which may be useful for future therapeutic development. It is difficult to develop a perfect model of any disease especially for those that are rare and present with a wide range of clinical symptoms. Thus, this work will be of great interest to a broad field of rare diseases. The reviewer appreciates the progress and novelty in this work, but the following issues should be addressed:

Major concerns

- No experimental evidence of myoclonic seizures is presented. In the discussion, the authors mention that they were “unable to determine whether P361R-SMA mice had any seizure activity.” Why is this the case? As PME is an important clinical phenotype of this disease, it seems crucial to determine if these mice have myoclonic epilepsy phenotypes.

We thank the Reviewer for this comment. In order to fulfill this directive, we initiated a new collaboration with a group that has expertise in these types of analyses. We evaluated seizure susceptibility of P361R-SMA mice by exposure to 10% flurothyl in accordance with a previously published protocol (DOI: [10.21769/BioProtoc.2309](https://doi.org/10.21769/BioProtoc.2309)). P361R-SMA mice exhibited a decreased threshold to, and an increased frequency of flurothyl-induced myoclonic jerks, which is a behavioral correlate of myoclonic seizures. Our data suggest that P361R-SMA mice are prone to the development of myoclonic epilepsy. These findings have been summarized in Supplementary Figure 3E-H, and described in the Results section as follows (inserted after line 257):

“To determine whether P361R-SMA mice exhibit an increased susceptibility to seizure development, we used the flurothyl model of repeated generalized seizure induction. Fifteen-week-old P361R-SMA mice and age-matched WT animals were subjected to eight flurothyl-induced seizures (once a day). Susceptibility to myoclonic jerks and generalized seizures were evaluated during a rechallenge of mice with flurothyl exposure after a 28 day rest period. We found a significant decrease in the flurothyl-induced myoclonic jerk threshold (Supplementary Fig. 3E) and a trend to an increased frequency of myoclonic jerks in P361R-SMA mice (Supplementary Fig. 3F). These findings suggest an increased susceptibility of this strain to myoclonic seizures. On the other hand, neither the threshold to a generalized seizure nor their frequency in response to flurothyl exposure was affected in this strain (Supplementary Fig. 3G,H).”

We have also modified text in the Discussion (lines 604-613) which now reads as follows:

“Myoclonic seizures are generally caused by perturbed electrical activity in the CNS, and are reported in some LSDs¹⁰¹. Myoclonic seizures can also occur in patients suffering from diseases with white matter degeneration¹⁰²⁻¹⁰⁶. In addition, numerous deficiencies affecting sphingolipid homeostasis or causing sphingolipid storage (besides sphingolipidoses already classified as leukodystrophies) have white matter abnormalities^{30-32,107-110}. Our data shows that P361R-SMA mice exhibit a decreased threshold and an increased frequency of flurothyl-induced myoclonic jerks in comparison with age-matched WT animals, which is a behavioral correlate of myoclonic seizures. These data suggest that P361R-SMA mice are prone to the development of myoclonic epilepsy. Further studies, including electromyography, are needed to determine overlap of clinical signs of SMA-PME with P361R-SMA mice.”

- The authors first make a T41A mouse based on another patient mutation. However, these mice are largely asymptomatic. The ACDase activity for this mutation needs to be evaluated (similar to Fig 1D,C). If this mutation also disrupts ACDase activity, it calls into question the conclusion that the lack ACDase activity is etiology behind SMA-PME.

We thank the Reviewer for the suggested experiment. We constructed a similar overexpression vector for the T41A mutation and repeated the enzyme assays. We have revised our Figure 1 to include our new data.

We do see reductions in enzyme activity with the T41A mutation in this HEK293T transfection assay. Further studies, that we believe are outside the scope of this present study, will be required to address the consequences of the P361R and T41A mutated ACDases in various cell types in vivo. Unfortunately, the fluorogenic enzyme activity assay we employ is unsuitable for mouse cells (previously discussed in Li et al 2019). Secondary evidence for a lack of enzyme activity in vivo is provided by the massive accumulation of ceramide; yet we were unable to demonstrate dramatic ceramide accumulation in T41A mice, which suggests that, despite our in vitro findings, this mutation does not reduce ACDase activity in vivo to a level that results in pathogenic outcomes.

Lastly, we would like to note that our primary reason for including data for T41A mice in the present manuscript was to save other investigators the effort of making this specific mouse strain, as T42A is a frequently occurring ACDase mutation related to SMA-PME.

- The histology images compare WT to P361R-SMA mice while the text continues to make comparisons to P361R-Farber mice from previous publications. Representative images of P361R mice need to be shown in the same panel as P361R-SMA mice so the reader can make their own comparisons without having to find previous references.

We thank the Reviewer for their suggestion. We agree that it would be difficult for a reader to easily access previously published work to make comparisons. However, we feel that inclusion of additional data for an already well-described model dilutes our story. As such, based on additional feedback from Reviewer 3, we have changed the text comparing P361R-SMA to P361R-Farber in Result section in relation to Figure 2 – originally lines 184-197. We have moved such comparisons of P361R-SMA to P361R-Farber mice into the Discussion for general comparison of both models (as suggested by Reviewer 3) after line 553.

- The lipid analyses in the current form are not extremely valuable to the reader and are somewhat a “data dump.” Discussion of lipid changes are largely presented as “lipid X changes in brain region Y” with minimal discussion of the biological relevance for these changes, especially ACVs. This reviewer suggests moving most of the lipid data to the supplement and highlighting only a handful of the most important lipid changes.

We thank the Reviewer for this suggestion and for the feedback. We have not included ACV changes as a major component of our main sphingolipid figures – we have only shown a summarized heatmap of ACV abundance. Changes in specific ACVs for each sphingolipid species are indicated in the supplementary data section. We have also not determined

sphingolipid changes in various brain or spinal cord regions, only in gross tissues, which is a shortcoming in our study that we have discussed. In addition, we would like to point the Reviewer's attention to our Discussion section where we have indicated the potential biological relevance of many of the changes in various species we have observed (lines 614-697, with discussion of what we deem to be important sphingolipid species – Cer, dhCer, DHCs, MHCs, THC_s, Sulfatides - as well as specific ACVs, for example C16:0, C18:0 and long-chain vs. short-chain). Finally, we would not be comfortable making strong conclusions about the role of each sphingolipid species without conducting detailed experiments to perturb each species individually; we believe such studies are outside the scope of this manuscript, but nonetheless important as a future direction. That said, we would not like to bias our readers in their own interpretation of our data and the importance of each sphingolipid, and would like to present similar information in a coherent format. As such, we feel that the current format best captures the complexity of the sphingolipid pathway and how it can be perturbed by a defect at a single point.

- In figure 3, ceramide is increased in both spleen and liver in the P361R-SMA mouse, as one would expect with disrupted ACDase activity. However, it is unclear why the product of this activity is also upregulated. Please explain.

We thank the Reviewer for this excellent question. Intuitively, the reviewer is right to expect that sphingosine levels ought to be reduced with an increase in ceramide levels. Indeed, this is what we have measured previously in P361R-Farber livers when the mice are in a mixed 129S6/B6 genetic background (see Yu et al. 2018 and Yu et al 2019). However, in brains of P361R-Farber mice with the same mixed genetic background we have - counterintuitively - measured an increase in sphingosine (see Sikora et al. 2017). We find the opposite results in both P361R-Farber and P361R-SMA mice in a pure C57BL/6J background as used in our study: sphingosine levels are reduced in brain and spinal cord alongside ceramide accumulation, but sphingosine levels are - counterintuitively - increased in liver and spleen. While the simplest explanation for this difference may be related to the different LC/MS methods used between the studies, the genetic background of the mice may also play a role. Thus, further studies are warranted to explore strain-specific differences in sphingolipid perturbation, as discussed (originally lines 647-657).

One possible explanation for the above observations, that we have postulated in our Discussion, is that extra-lysosomal ceramidases, for example ACER1-3 and ASAH2, may act on the increased ceramide substrate available, resulting in an aberrant albeit minor increase relative to other sphingolipids that accumulate. The degree of expression of these ceramidases, or leakage of ceramide accumulated in the lysosome into other cellular compartments may be strain specific. However, as we do not expect the significant but minor increase in sphingosine to participate significantly in SMA-PME pathogenesis, we consider these important metabolic studies outside the scope of the current manuscript.

Minor

- In figure 3, "Spleen" overlaps with other text on the left side of the figure (near B)

We thank the Reviewer for pointing out this formatting error, which has now been corrected.

- Need to label the left side of the bottom panel of figures in Fig 4E.

We thank the Reviewer for this suggestion and have labelled the various panels in Fig. 4E.

Reviewer #3 (Remarks to the Author):

In this manuscript entitled “Unique Spinal Muscular Atrophy-Like Phenotype in a Novel Mouse Model of Acid Ceramidase Deficiency”, the authors show that a mouse model with spinal muscular atrophy with progressive myoclonic epilepsy (SMA-PME)- like phenotype (P361R-SMA). Although most of the data in this manuscript are convincing and presented by well-designed experiment, I have listed my concerns below:

Major comments

1. Is there any effect of sex difference on the pathological condition? In Figure 1, only weight is shown as males and females, but what about other conditions? Please describe the sex used in the experiment. If you're using both genders, it might be a good idea to color-code each data plot sex.

We thank the Reviewer for their vigilance and their helpful comment. Per the Reviewer’s suggestion, we have indicated sex information for individual data points by color coding or using different symbols. We used both sexes for most of our studies, with at least n=3 of each sex. Before pooling data points as shown, statistical comparisons were made between values obtained for both sexes. No differences were seen, except for weight; and this can be attributed to expected and well documented differences in weight between male and female mice. Specifically, we have modified the following figures: Fig. 1F, Fig. 2D-I, Fig. 3, Fig. 5, Fig. 7, Supplementary Fig. 3A-H, and Supplementary Fig. 4 to indicate individual data points and sex where appropriate. Low resolution images are inserted below:

Also, I recommend describing the effects of gender differences in this mouse model and sex differences in general epidemiology in the discussion section.

We thank the Reviewer for this recommendation and have added a statement (at line 513) in our discussion to indicate that sex-specific differences have not been reported in SMA-PME patients, but that available data is too sparse to make an accurate conclusion:

“No sex-specific differences have been reported to our knowledge, although clinical data is too sparse to accurately make a conclusive statement.”

We have also added a statement to indicate that we have not observed any sex-specific differences in our studies, in general (at line 541):

“In addition, no difference in the presentation of physiological or biochemical phenotypes were observed between male and female P361R-SMA mice.”

Minor comments

1. In Figure 1H, It is difficult to understand which organ it is, please add explanations. In the Figure legend, the authors explain that "Mice and organs were placed next to a ruler for accurately scaled resizing of images, such that WT and P361R-SMA images can be directly compared", but it is more accurate to insert the scale bar.

We thank the Reviewer for their suggestion and have added labels for the various tissues shown, as well as a scale bar.

2. In Lines 184-191, Although there is a description to compare the pathology of P361R-SMA mice to P361R-Farber mice, the actual figure is a comparison of Control and P361R-SMA mice. Although this may be the case when compared with previous studies, It may not be appropriate to compare P361R-SMA mice and P361R-Farber mice as result. As shown in the data & figure presented, it would be better to describe that MAC-2 has increased compared to control, and compare P361R-Farber mice of past results in the discussion sections. If it is necessary to compare the pathology of P361R-SMA and P361R-Farber mice, please present the data and perform statistical comparisons.

We thank the Reviewer for their suggestion. We agree with the Reviewer, and we have modified the text comparing P361R-SMA with WT in the Results section describing Figure 2 (originally lines 184-197). We have moved the paragraph comparing P361R-SMA to P361R-Farber mice into the Discussion (after line 553).

3. In Figure 2D, you may need to present an image of the Western blot. Also, please show the data of each sample with dot plot like other data.

We thank the Reviewer for identifying the inconsistencies in our data presentation. We have added images of representative immunoblots and reworked the graphs as suggested (see inserted below).

4. In the lines 394-395 and Figure 6D, although CatD+ cluster is overlapped with Mac-2 in the serial section, the parts that do not overlap are more visible. It would be better to quantify the percentage of overlapping cells.

We thank the Reviewer for this suggestion. We have removed the sentence describing the overlap of CatD+ staining with Mac-2 for Figure 6D from our manuscript. We have been working on optimizing our antibodies for alternative staining methods to better present these findings and be able to quantify them in the future. Unfortunately, we are not able to present quantitative data at this point, though we believe it is not imperative to accurately describe our new model.

Reviewers' comments:

Reviewer #1 (Remarks to the Author):

The authors have not adequately attended to address the reviewer concerns particularly around better description of organ atrophy (Fig 1H) – it is insufficient to simply address the concerns in the rebuttal without any additional data or points added in the revised paper. Please incorporate data and discussion points in the paper proper, not simply a rebuttal for the reviewer.

Figure 1H which attempts to address the reviewer concern about the poor representation of the atrophied organs are still not up to standards of scientific data presentation.

Removing the data to address significant concerns around histology to show atrophy of the tissues is not an acceptable way to address reviewer comments. This point was indeed raised by all three reviewers and to omit the data altogether suggesting that this dilutes their paper is not acceptable.

Reviewer #3 (Remarks to the Author):

All the points raised in the first version of the manuscript have been properly addressed by the authors.

Response to reviewers

Please see responses in red text.

Referee expertise:

Referee #1: Mouse models of vascular and metabolic disorders

Referee #3: Mouse models of metabolic and neurological disorders

Reviewer #1 (Remarks to the Author):

The authors have not adequately attended to address the reviewer concerns particularly around better description of organ atrophy (Fig 1H) – it is insufficient to simply address the concerns in the rebuttal without any additional data or points added in the revised paper. Please incorporate data and discussion points in the paper proper, not simply a rebuttal for the reviewer.

We thank the Reviewer for their feedback, and the suggestion to include more of the rebuttal comments regarding Fig. 1H in the manuscript. We have incorporated the comments in question into our Discussion as follows:

... “Visceral organs were atrophied in P361R-SMA mice, which we hypothesize is proportional to the loss in body weight observed, as with mouse models of other sphingolipidoses^{49,58}. Organomegaly, observed in P361R-Farber mice^{36,38}, has been postulated to occur due to massive accumulation of foamy macrophages with storage material, which is not seen in P361R-SMA mice.”

Figure 1H which attempts to address the reviewer concern about the poor representation of the atrophied organs are still not up to standards of scientific data presentation.

We appreciate the additional feedback from the Reviewer regarding our Figure 1H. The Reviewer previously made the following comment:

“Figure 1H needs better labeling and description. The images are not at all clear based on how they are laid out.”

To this end, we labelled the various tissues. Based on feedback from Reviewer #3 and the Editor, we feel that our figure is sufficient to showcase what key visceral organs look like and how their sizes compare to those from WT mice. We have deferred to an Editorial decision regarding this comment from the Reviewer and have not made any further modifications.

Removing the data to address significant concerns around histology to show atrophy of the tissues is not an acceptable way to address reviewer comments. This point was indeed raised by all three reviewers and to omit the data altogether suggesting that this dilutes their paper is not acceptable.

We thank the reviewer for their feedback. As noted to the Editor, the histology we originally presented as Supplementary Fig. 3 was re-reviewed by a specialist who advised us that the

images were not comparable. We have added back into the manuscript a more appropriate comparison of similarly oriented muscle tissue. We note that our aim with the original figure was to investigate myofiber atrophy, as may be expected in SMA-like states, not general organ atrophy. Our current figure addresses this question to a degree we feel is sufficient for the scope of this manuscript.

The following text was added to the Results section:

“We first evaluated skeletal muscle histopathology in several mice. Comparison of similarly oriented areas of muscle identified a subpopulation of myofibers in P361R-SMA that were smaller than the corresponding population in WT mice, suggesting mild myofiber atrophy (Supplementary Fig. 3), although it should be noted that sampling error may be responsible for this apparent difference.”

Reviewer #3 (Remarks to the Author):

All the points raised in the first version of the manuscript have been properly addressed by the authors.

We thank the Reviewer for their positive feedback.